# Quantitative CRACI reveals transcriptome-wide distribution of RNA dihydrouridine at base resolution

Cheng-Wei Ju [1,2,3,13], Han Li [1,2,13], Bochen Jiang [1,2,4], Xuanhao Zhu[1,2], Liang Cui [5], Zhanghui Han[6], Junxi Zou[1,2], Yunzheng Liu[7], Shenghai Shen[8,9], Hardik Shah [10], Chang Ye [1,2], Yuhao Zhong[1,2], Ruiqi Ge [1,2], Peng Xia [1,2], Yiyi Ji [1,2], Shun Liu [1,2], Fan Yang[1,2], Bei Liu[1,2], Yuzhi Xu[11], Jiangbo Wei [6,12], Li-Sheng Zhang [8,9] ✉ & Chuan He [1,2,3] ✉

Dihydrouridine (D) is an abundant RNA modification, yet its roles in mammals remain poorly understood due to limited detection methods. We even do not have a comprehensive profile of D site location and modification stoichiometry in tRNA. Here, we introduce **C**hemical **R**eduction **A**ssisted **C**ytosine **I**ncorporation sequencing (CRACI), a highly sensitive, quantitative approach for mapping D at single-base resolution. Using CRACI, we generate the transcriptome-wide maps of D in both cytoplasmic and mitochondrial tRNAs from mammals and plants. We uncover D sites in mitochondrial tRNAs and identify DUS2L as the 'writer' protein responsible for human mitochondrial tRNAs. Furthermore, we demonstrate that most D modifications have a limited impact on tRNA stability, except for D20a, which also exhibits cis-regulation of adjacent D20 sites. Application of CRACI to human mRNA reveals that D modifications are present but rare and occur at very low stoichiometry. CRACI thus provides a powerful platform for investigating D biology across species.

Posttranscriptional modifications in RNA molecules from various species play crucial roles in regulating gene expression. Among the over 150 chemically modified nucleosides identified in ribosomal RNA (rRNA), transfer RNA (tRNA), messenger RNA (mRNA), and other non-coding RNA (ncRNA), dihydrouridine (D) is one of the most prevalent modifications found in tRNAs across bacteria, eukaryotes, plants, and some archaea[1–3]. D results from the reduction of the carbon-carbon double bond at positions 5 and 6 of the uridine, yielding a fully saturated pyrimidine ring[4,5]. Uridine modifications are prevalent in tRNA molecules[6]. D is the second most abundant modified nucleoside found in tRNAs, predominantly installed in the D-loop of tRNAs for which it is named, only behind pseudouridine (Ψ) long recognized as the most abundant modification in tRNAs[2,7]. The conversion of uridine to D is catalyzed by DUS enzymes (dihydrouridine synthases) using FMN and NADPH/NADH as a cofactor[2,8]. The presence of all three Dus enzymes (DusA, DusB, DusC) were observed in Proteobacteria, while

[1]Department of Chemistry, The University of Chicago, Chicago, IL, USA. [2]Howard Hughes Medical Institute, The University of Chicago, Chicago, IL, USA. [3]Pritzker School of Molecular Engineering, The University of Chicago, Chicago, IL, USA. [4]School of Life Sciences & Biotechnology, Shanghai Jiao Tong University, Shanghai, China. [5]Antimicrobial Resistance Interdisciplinary Research Group, Singapore-MIT Alliance for Research and Technology, Singapore, Singapore. [6]Department of Chemistry, National University of Singapore, Singapore, Singapore. [7]Division of Biology and Biological Engineering, California Institute of Technology, Pasadena, CA, USA. [8]Division of Life Science, The Hong Kong University of Science and Technology, Kowloon, Hong Kong SAR, China. [9]Department of Chemistry, The Hong Kong University of Science and Technology, Kowloon, Hong Kong SAR, China. [10]Metabolomics Platform, Comprehensive Cancer Center, The University of Chicago, Chicago, IL, USA. [11]Department of Chemistry, New York University, New York, NY, USA. [12]Department of Biological Sciences, National University of Singapore, Singapore, Singapore. [13]These authors contributed equally: Cheng-Wei Ju, Han Li. ✉e-mail: zhangls@ust.hk; chuanhe@uchicago.edu

Gram-positive bacteria only possess DusB[9,10]. Four Dus enzymes exist in budding yeast (Dus1, Dus2, Dus3, Dus4)[10]. Dus1 and Dus4 in yeast are dual-site enzymes that catalyze the formation of D16/D17 and D20a/D20b, respectively, whereas Dus2 and Dus3 are specific to a single site, synthesizing D20 and D47, respectively[11,12]. More recently, RdsA has been identified as the ribosomal dihydrouridine synthase for D2449 in *E. coli* 23S rRNA[13]. In mammalian cells, four homologs of the yeast Dus are characterized as DUS1L, DUS2L, DUS3L, and DUS4L[14]. However, the functional roles of these mammalian DUS enzymes in modifying D sites on tRNA or mRNA have yet to be fully established.

Structurally, D is unique since it removes the aromaticity of the uracil ring, therefore exhibiting the potential to impact the intramolecular base-stacking within an RNA molecule, intermolecular base-pairing, and RNA folding[15]. This unique structural feature of D promotes RNA flexibility and further destabilizes the C3′-endoribose conformation associated with base-stacked RNA[4,5]. Besides the direct effect of D on RNA flexibility and local folding, D has recently been shown to also affect mRNA translation and involved in human diseases. *DUS3L*-knockout (KO) cells have compromised protein translation rates and impaired cellular proliferation[14]. Also, it has been known that D levels are increased in non–small cell lung carcinomas (NSCLC), which correlates with *DUS2L* overexpression with DUS depletion suppressing tumor growth[16]. Additionally, DUS1L is also identified as the top important candidate among epitranscriptomic regulators in the initiation and metastatic transformation of colorectal cancer (CRC)[17]. A recent study identified DUS1L as the dihydrouridine synthase responsible for D16/D17 in human tRNAs, where *DUS1L* overexpression impairs tRNA processing and translation in glioblastoma[18].

As the two predominant uridine modifications in mammalian cells, the functional investigation of Ψ and D has long been limited by the lack of sensitive and quantitative mapping tools. In contrast, research on $N^6$-methyladenosine (m6A) in mRNA and non-coding RNAs has rapidly advanced, due to NGS technology, which enables comprehensive profiling of m6A distributions across the transcriptome[19–23]. We have been actively studying uridine modifications in various RNA species; we made significant progress by developing BID-seq, which allows for base-resolution quantitative sequencing of Ψ[24–28]. BID-seq utilizes highly specific chemical conversion at Ψ-modified sites, followed by the analysis of deletion signatures to quantify Ψ dynamics and study Ψ biology. Several sequencing methods for detecting D modification have also been developed, including AlkAniline-Seq, Rho-seq and D-seq[29–32]. These methods rely on RT stop, which limits their sensitivity and poses challenges in detecting D in heavily modified regions of small RNA species, such as tRNAs[33–35]. We aim to achieve quantitative sequencing of D modifications across the transcriptome at single-base resolution with: (1) quantitative sequencing revealing D modification stoichiometry at each modified site, allowing the interpretation of D dynamic profiles through misincorporation or deletion signatures; (2) monitoring dynamic changes in D modification stoichiometry in response to various cellular perturbations, such as gene knockdown, heat shock, and hypoxia, expanding our understanding of how D 'writer' proteins (DUS) regulate specific D deposition; (3) providing insights into the differing or unified behaviors of D 'writer' proteins across multiple species; (4) confirming the presence of D modifications in non-tRNA species, including mRNA, lncRNA, and repeat RNAs, laying the groundwork for future studies on D biology in various biological and physiological processes.

Here, we present Chemical Reduction Assisted Cytosine Incorporation sequencing (CRACI) for whole-transcriptome quantitative mapping of D modifications at single-base resolution, demonstrating high sensitivity in uncovering D-modified sites with high, moderate, or low stoichiometry in mammals and plants.

## Results

### CRACI maps D as internal misincorporation signatures

After the removal of the uracil ring aromaticity in the D base, the fully saturated structure can be further reduced using strong reduction reagents like KBH4 to produce the reduced D (Fig. 1a). The chemical reduction approach was previously introduced in the study of $N^4$-acetylcytidine (ac4C), enabling precise identification and quantification of ac4C in multiple RNA species[36–38]. In the presence of common reverse transcriptases, reduced dihydrouridine (D) generated through chemical reduction typically triggers RT truncations, complicating D detection in D-seq and Rho-seq (Fig. 1b)[30–32]. This hinders more accurate measurement of D modification stoichiometry and causes high background noise, particularly for lowly modified D sites in mammalian mRNA. Furthermore, RT truncation signatures can prevent reverse transcriptase from reading subsequent D sites when multiple D modifications are densely clustered within a narrow RNA region (Fig. 1c). To address these challenges, we developed CRACI to map D modifications as internal misincorporation signatures, providing stoichiometry information, reducing background noise, and enabling the detection of densely-clustered D sites (Fig. 1b).

We reanalyzed tRNA-seq data using various commercially available RTs, including SuperScript II, SuperScript III, SuperScript IV, HIV RT, MMLV RT, RT-1306, RT-41B4, and TGIRT[23,39,40]. We found that SuperScript IV RT (SSIV RT) and HIV RT could induce mutation at D sites on RNA. Our subsequent experiments detected approximately 3% and 12% weak misincorporation ratios at dihydrouridine (D) sites, respectively, based on the average misincorporation ratio of 256 motifs in the NNDNN context (Fig. 1c). Although these misincorporation ratios were not particularly high, the patterns were predominantly T→C mutations, with few observable T→A or T→G mutations. We have achieved higher mutation ratio for 2′-O-methylation (Nm) detection by adjusting the dNTP/dATP ratios in our previously Nm-Mut-seq work[39,40]. We speculated that systematically adjusting the dNTP/dGTP ratios during RT reactions might increase T→C misincorporations at D when using HIV RT. Using a mixture of 1 mM dGTP and 10 μM dNTP for RT, we detected an average T→C mutation rate of approximately 20% at D with HIV RT. Further optimization of this 1 mM/10 μM dGTP/dNTP condition and apply an optimized 3-hour KBH4 reduction of D achieved an average T→C mutation rate of around 96% at the reduced D sites (Fig. 1c). Liquid chromatography-tandem mass spectrometry (LC-MS/MS) analysis confirmed the near-quantitative formation of tetrahydrouridine, without detectable ureidopropional products under our KBH4 reduction treatment condition (Supplementary Fig. 1a). The KBH4 reduction renders D less planar and destabilizes U−A pairing, shifting it to mispair with G (Supplementary Fig. 1b). The permissive readthrough by HIV reverse transcriptase, in the presence of elevated dGTP/dNTP ratio, further enhances T→C misincorporation at the reduced D sites. Subsequent adjustments of the dGTP/dNTP ratios at 1 mM/20 μM, 1 mM/50 μM, or 1 mM/100 μM yielded similar mutation rates at the reduced D using HIV RT (Supplementary Fig. 1a). Interestingly, testing the dGTP/dNTP ratios in HepG2 cellular small RNA revealed an increased yield of final libraries, indicating improved read-through ability of HIV RT in the presence of higher dGTP/dNTP ratios (Supplementary Fig. 1b). Consequently, we established Chemical Reduction Assisted Cytosine Incorporation sequencing (CRACI) using a 1 mM/100 μM dGTP/dNTP ratio to enhance RT read-through, enabling whole-transcriptome quantitative mapping of D modifications after reduction at base resolution.

### CRACI enables quantification of D stoichiometry

Using CRACI, the analysis of T→C mutation signatures across all 256 motif contexts in NNDNN reveals extremely high mutation rates, ranging from 91% to 99%. This indicates the use of HIV-RT in CRACI does not lead to significant sequence context bias around D sites (Fig. 1d). In contrast, while SSIV-RT also exhibits high mutation rates, the presence

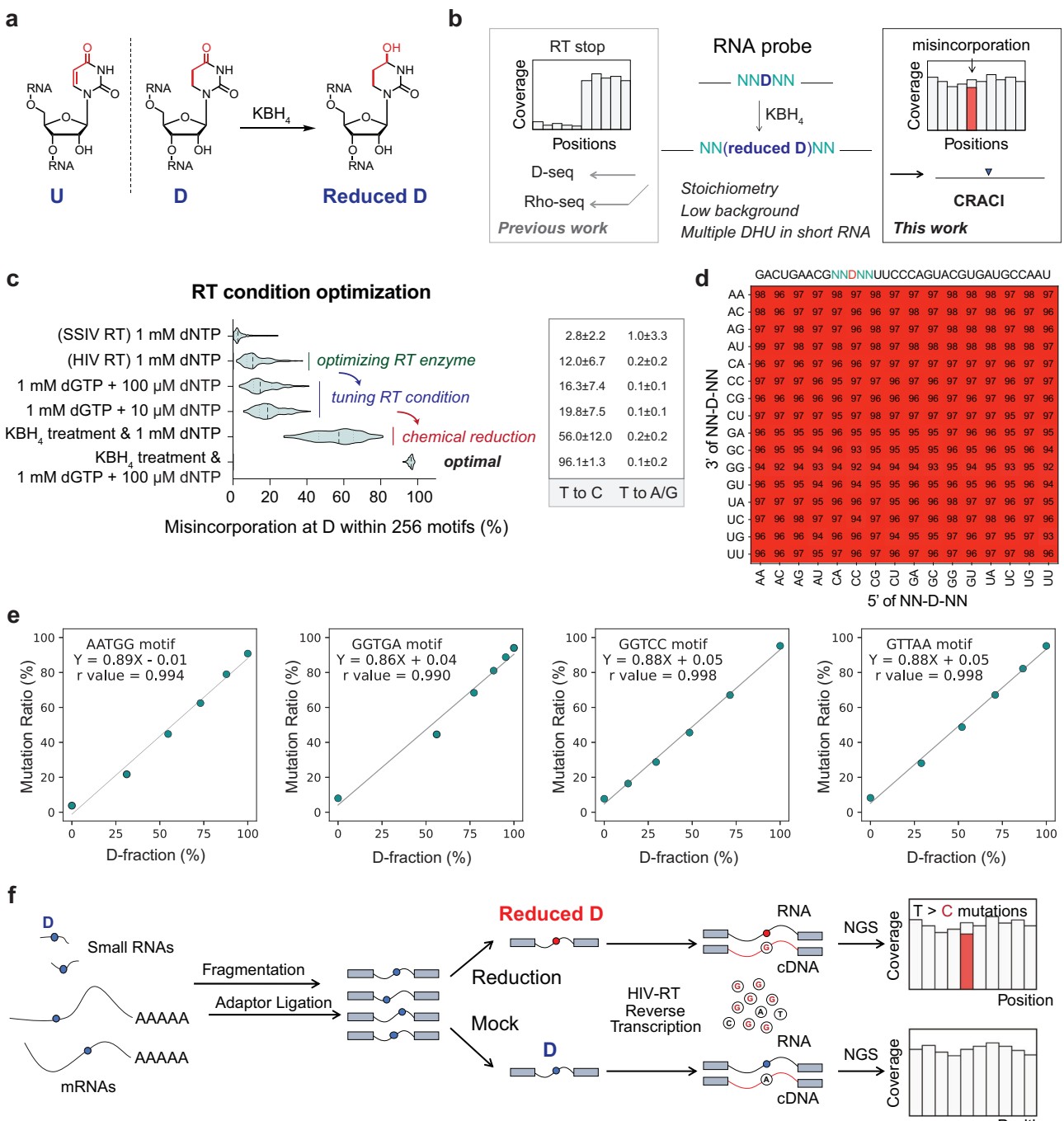

**Fig. 1 | Development of CRACI for base-resolution quantitative sequencing of D. a** Schematic plot of the chemical principle underlying the reduction of D by KBH₄ treatment. **b** Comparative analysis of RT-truncation-based sequencing methods versus misincorporation-based methods for D detection. **c** Optimization of RT conditions increases misincorporation at D sites, shown by the misincorporation ratios of 256 NNDNN motifs. The violin plot (right) shows misincorporation distributions across 256 motifs under different conditions, with median (dashed line), interquartile range (dotted lines), data density (width), and extremes. Mutation patterns (T to C or T to A/G) are summarized as mean ± SD across 256 motifs. **d** Motif-dependent misincorporation patterns observed in CRACI using a synthetic 35-mer RNA oligo centered by NNDNN, in the presence of HIV RT and 1 mM/100 μM dGTP/dNTP combination. **e** Representative sequence-context-dependent calibration curves (misincorporation ratio vs. D fraction) for D stoichiometry quantification using CRACI. **f** A flowchart of CRACI library preparation, detecting D fractions through T-to-C misincorporation signatures.

of motif context bias limits its applicability in cellular systems (Supplementary Fig. 1c). Additionally, unmodified probes containing 0% D (NNUNN) showed mutation rates of less than 4-15% across all sequence motifs (95% CI: 7.2%, 7.7%; Supplementary Fig. 1d), which can be effectively filtered out using established D detection criteria in the analysis pipeline. To demonstrate the quantitative capabilities of CRACI, we constructed libraries using a mixture of synthetic RNA

oligos containing NNDNN or NNUNN to create calibration curves for each motif. The mild CRACI KBH₄ reduction treatment, which avoids RNA degradation, and the enhanced RT read-through at reduced D in the presence of HIV RT when using 1 mM/100 μM dGTP/dNTP combination allowed us to achieve mostly linear calibration curves for all 256 motif contexts in NNDNN (Fig. 1e, Supplementary Fig. 1e and Supplementary Data 1), rarely seen in such methods and enabling truly

quantitative mapping of D in most RNA species. To evaluate potential false-positives in CRACI, we analyzed mutation profiles across cellular 18S and 28S rRNA in treated samples. No significant increases in canonical base mutation rates (based on the defined cut-offs) nor any notable RT truncation signals were observed (Supplementary Fig. 2). Starting with purified polyA+ RNA or small RNA (<200 nt) from various biological samples, we therefore developed a comprehensive CRACI protocol to produce high-quality NGS libraries, with the steps of RNA fragmentation, end repair, 3'-/5'-adaptor ligation, reverse transcription, and PCR amplification (Fig. 1f). This strategy of adaptor ligation combined with UMI has been broadly applied in previous studies of small RNAs and miRNAs, showing low bias, high reproducibility, and effective PCR duplicate removal[41,42].

## Quantitative CRACI uncovers D in human cytoplasmic and mitochondrial tRNAs

D modifications are predominantly located in the D-loop of cytoplasmic tRNAs in mammalian cells. To investigate this, we isolated cellular small RNAs (<200 nt length) from cultured HepG2 cells and conducted CRACI analysis. Our findings revealed that D modifications are specifically present at positions 16, 17, 20, 20a, 20b, and 47 in HepG2 cytoplasmic tRNAs (ct-tRNAs), with no D signatures detected at other positions of tRNA (Fig. 2a, b and Supplementary Data 2). The readthrough capability of HIV RT at methylated or modified sites within tRNAs allowed CRACI to identify multiple D modifications in fragmented tRNAs. Across 46 different cytoplasmic tRNAs from HepG2 cells, we comprehensively mapped D modifications using CRACI, providing detailed quantification information (Fig. 2c and Supplementary Fig. 3a). To the best of our knowledge, this represents the first quantitative atlas of D profile in this context. The observed T to C mutation ratios were converted into D modification stoichiometry using our calibration curves. Notably, certain human tRNAs, such as Ile-UAU and Ala-CGC, exhibit a single D-modified site, while others like Arg-UCU and Phe-GAA are modified by D at multiple positions (Fig. 2c and Supplementary Fig. 3b)[43,44]. Pooling D sites from all human tRNAs, we found that D16 and D17 are highly modified (>70% stoichiometry), D20 and D47 predominantly exhibit stoichiometry above 60%, and D20a shows a range of D stoichiometry from 20% to 100% (Fig. 2d). The absence of correlation between tRNA expression levels and detected D sites confirms that the observed D modifications are unlikely affected by sequencing coverage (Supplementary Fig. 3c).

To validate CRACI, we compared 10 previously reported D sites in tRNAs known as DUS2L in vitro substrates[45]. We detected 8 of these sites in our CRACI analysis (Fig. 2e and Supplementary Fig. 3d), while the remaining 2 sites did not meet the criteria for D detection as they were acp³U modifications[46] (Supplementary Fig. 3e). The DUS1L-regulated D16/D17 sites and DUS2L-regulated D20 sites identified through 5-ClUrd-iCLIP show strong overlap with the D16/D17/D20 sites detected by CRACI, with the top 10 iCLIP sites perfectly matching the CRACI results (Fig. 2f, g)[47]. Similarly, the DUS3L-regulated D47 sites identified by 5-FUrd-iCLIP also demonstrate excellent concordance with the CRACI findings (Fig. 2h)[14]. CRACI also identified several D sites not captured by CLIP experiments (Supplementary Fig. 3f–h). Additionally, the overlap between our data and Modomics further supports the reliability of CRACI-based D detection[48]. Although Modomics annotated only 30 D sites, most with low confidence (score 5, indicating "Evidence not yet annotated [Unknown]"), CRACI successfully detected and verified 27 of them (Supplementary Fig. 3i). This strong concordance underscores the robustness and sensitivity of CRACI. The three Modomics sites not detected by CRACI are detailed in Supplementary Fig. 3j: (1) tRNA-His-GTG D16, where no detectable D modification was observed in HepG2 cells; (2) tRNA-Met-CAT D20, where high mutation ratios were seen in both CRACI-treated and input samples, likely due to other uridine modifications such as acp³U or acp³D; and (3) tRNA-Tyr-GTA D16, with similar high background

mutation ratios possibly caused by non-D modifications. Together, these findings demonstrate CRACI's high accuracy and confidence in transcriptome-wide D detection.

In addition to cytoplasmic tRNA D modifications, we analyzed D signatures in HepG2 mitochondrial tRNA. CRACI only detected D signals at positions 16, 17, and 20 of mt-tRNAs (Fig. 2i). CRACI revealed five D-modified sites in mt-tRNAs: mt-tRNA^Asn D17, mt-tRNA^Asn D20, mt-tRNA^Gln D16, mt-tRNA^Gln D20, and mt-tRNA^Leu(UUR) D20. Among these, three highly modified D sites (above 90% stoichiometry) were confirmed by mass spectrometry[49], while the other two D sites (mt-tRNA^Asn D17 and mt-tRNA^Gln D16) were newly identified with moderate D stoichiometry (below 40% D fraction), highlighting the sensitivity of CRACI in D detection (Fig. 2j, k and Supplementary Fig. 4a). LC-MS/MS analysis of mt-tRNAs enriched by pulldown using single strand DNA probes further supported these newly identified mitochondrial D sites in HepG2 cells (Supplementary Fig. 4b).

Although D has been reported in yeast snoRNA[32], our CRACI results suggest the absence of D in other ncRNAs, such as snRNAs, snoRNAs, and other small ncRNAs, besides mt-tRNAs and ct-tRNAs, in the human cell line (Supplementary Fig. 4c–e).

## CRACI assigns the 'writer' proteins to D modifications in human cytoplasmic and mitochondrial tRNAs

The four homologs of yeast Dus, DUS1L, DUS2L, DUS3L, and DUS4L, have been characterized in mammalian cells. To investigate whether these dihydrouridine synthase (DUS) proteins independently regulate one or multiple D depositions or interact with each other for D installation, we depleted DUS1L, DUS2L, DUS3L, and DUS4L in HepG2 cells using siRNA knockdown. Following this, we purified small RNAs (<200 nt) and performed CRACI analysis. For D modifications in HepG2 cytoplasmic tRNAs, CRACI quantitatively monitored changes in D stoichiometry at each D site upon DUS depletion and precisely identified their cellular substrates (Fig. 3a and Supplementary Fig. 5a): DUS1L installs D16 and D17, DUS2L installs D20, DUS3L installs D47, and DUS4L installs D20a (Fig. 3b–e, Supplementary Fig. 5b–e and Supplementary Data 3). Interestingly, similar to NSUNs and PUSs, which act as 'writer' proteins for m⁵C and pseudouridine in tRNAs, we observed a compensatory effect: the levels of other D modifications increased when one DUS was depleted (Fig. 3b–e). Our results also suggest that reduced D levels may initially affect a specific subgroup of tRNAs rather than all tRNAs equally (Supplementary Fig. 5b–e). At least 70% of D sites exhibited reduced modification levels following simultaneous knockdown of all four DUS enzymes (Supplementary Fig. 5f). However, since we performed siRNA-based transient knockdown, residual DUS protein and the long half-life of tRNAs may have allowed certain D sites to remain at high modification levels[39]. Thus, we cannot exclude the possibility of functional redundancy among human DUS enzymes. Furthermore, in HepG2 mitochondrial tRNAs, we identified DUS2L as the writer for D deposition at D16, D17, and D20. This finding validated newly identified D sites and suggested DUS2L as an active 'writer' protein responsible for D modifications inside mitochondria (Fig. 3f, g).

Subsequent RNA-seq analysis of cytoplasmic tRNA abundance revealed that only DUS4L depletion led to a statistically significant decrease in the RNA abundance of its tRNA substrates, when compared to tRNAs not modified by D at position 20a (Fig. 3h and Supplementary Fig. 6a–d). This finding suggests that the D20a modification may regulate the stability of cytoplasmic tRNAs (as discussed in more details below).

## CRACI reveals the interaction and co-regulation of adjacent D modifications within the D-loop

Notably, quantitative CRACI can track interactions among multiple D sites within the D-loop, where four DUS enzymes target different sites. Perturbing one DUS enzyme can alter D stoichiometry at a specific site and may impact nearby D deposition. Analysis of CRACI data from

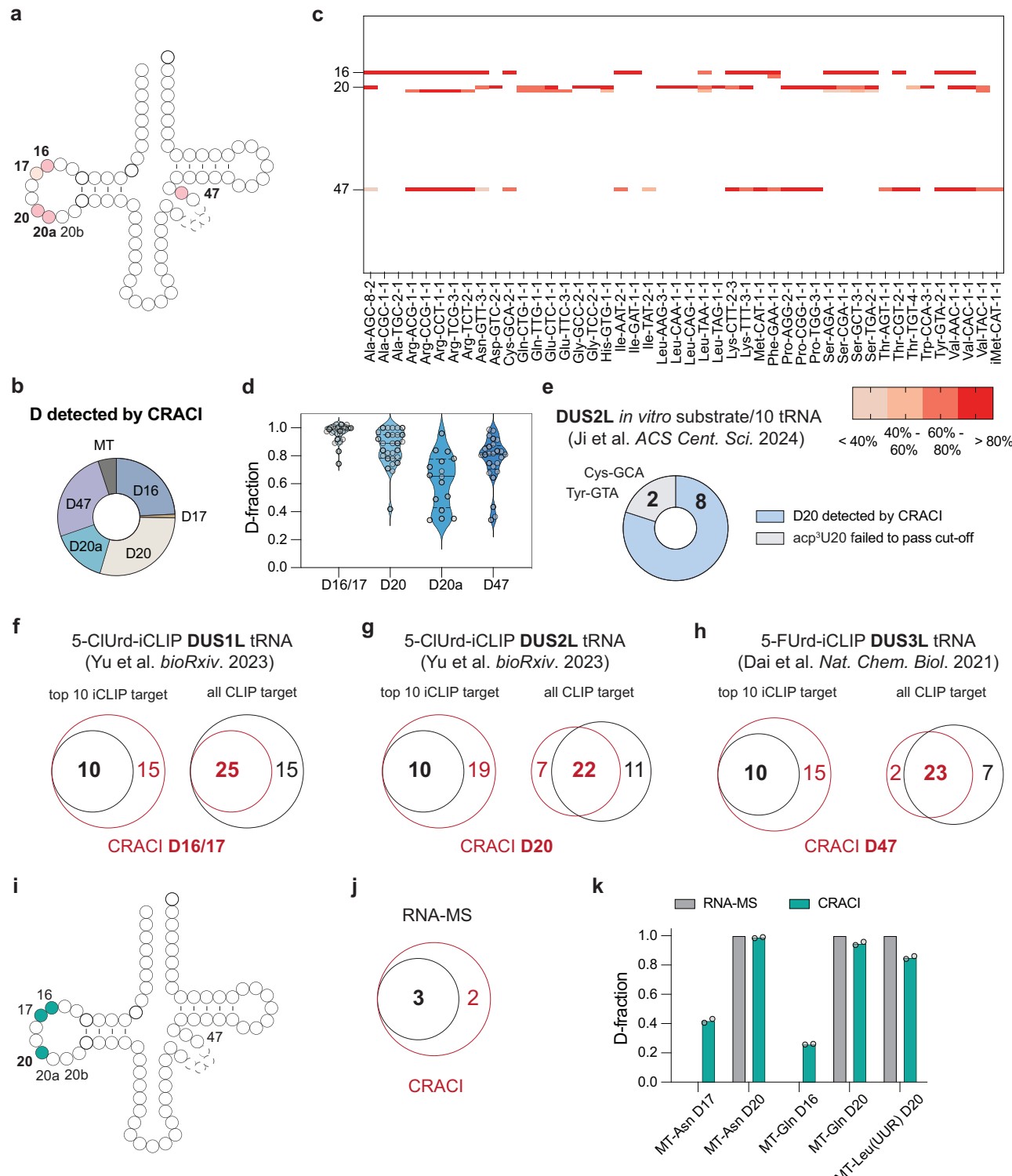

Fig. 2 | CRACI profiles D sites in HepG2 tRNA in a quantitative manner. a The identified D sites in cytoplasmic tRNAs (ct-tRNAs) from G2 cells, marked in pink color. b The number of D sites identified by CRACI in HepG2 ct-tRNAs, which are located at positions 16, 17, 20, 20a, and 47. c Heatmap displaying D modification stoichiometry at high-confidence D sites in HepG2 ct-tRNAs. A representative tRNA isoform in each tRNA class is shown. The mutation ratio was calculated as the average of two biological replicates. d Comparative analysis of D stoichiometry at positions 16/17, 20, 20a, and 47 in HepG2 ct-tRNAs, with violin plots illustrating the distribution of modification stoichiometry. Each point was calculated as the average of two biological replicates. e CRACI detected eight D20 sites that overlap very well with the 10 previously reported D20 sites installed by DUS2L[45]. The remaining two tRNAs harbor not only D but also acp³U at position 20, resulting in mutation

signatures even without treatment. Venn diagrams illustrating the overlap of CRACI-detected D sites with 5-ClUrd-iCLIP targets in human ct-tRNAs at f D16/17 and g D20 sites, installed by 'writer' proteins DUS1L and DUS2L, respectively[47]. h Venn diagram showing the overlap of CRACI-detected D47 sites with 5-FUrd-iCLIP targets in human ct-tRNAs, corresponding to the 'writer' protein DUS3L[14]. i Identification of D sites by CRACI in mitochondrial tRNAs (mt-tRNAs) from human HepG2 cells. j The overlap of CRACI-detected mt-tRNA D sites with a previously published mass spectrometry dataset[49]. k Quantitative observation of mt-tRNA D sites by CRACI, demonstrating the concordance with mass spec data. Two biological replicates are used for CRACI and shown here. The quantitative D ratios for RNA-MS were obtained from a published mass spectrometry dataset[49].

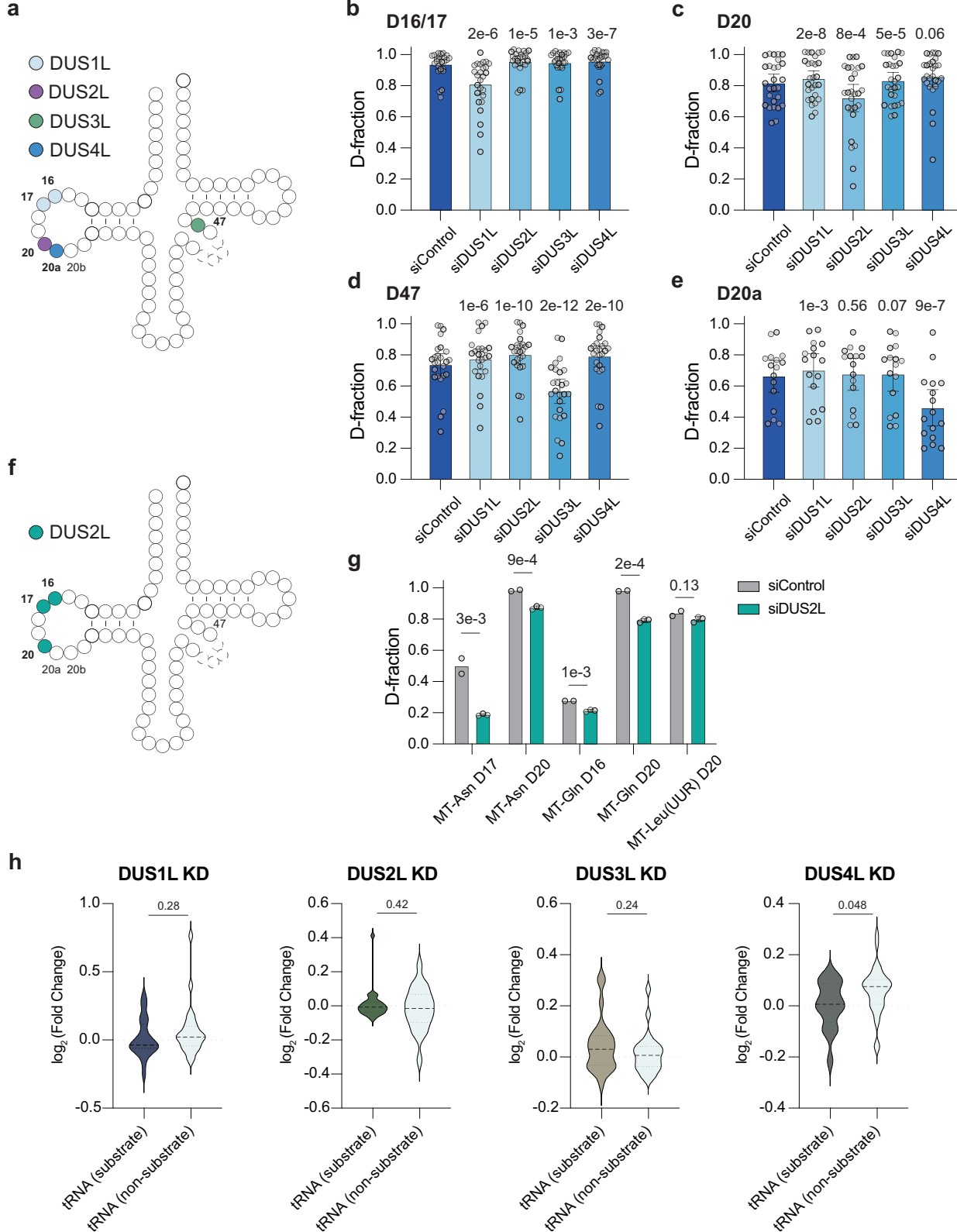

**Fig. 3 | CRACI assigns 'writer' protein to each D-modified site in human tRNA.**
**a** An integrated overview of DUS-dependent D profiles in HepG2 ct-tRNAs. Comparative analysis of D stoichiometry dynamics at specific sites within HepG2 ct-tRNAs following 72-hour siRNA-mediated knockdown of 4 DUS enzymes: **b** D16/17 (n = 26), **c** D20 (n = 27), **d** D47 (n = 25), and **e** D20a (n = 16). *P*-values from two-sided paired t-tests are displayed. Each point was calculated as the average of at least two biological replicates. **f** Integrated overview of DUS2L-regulated D sites in HepG2 mt-tRNAs. **g** Comparison of D modification levels in mt-tRNAs in *DUS2L* knockdown

versus the control. Two biological replicates are used for control samples and three biological replicates are used for DUS2L knockdown samples. P-values from two-sided unpaired t-tests are displayed. **h** Comparison of tRNA expression levels in response to depletion of 4 DUS enzymes, with tRNAs categorized into DUS-targets versus non-DUS targets. Expression levels of different RNA species were calculated as the average of at least two biological replicates. P-values from two-sided unpaired t-tests are displayed.

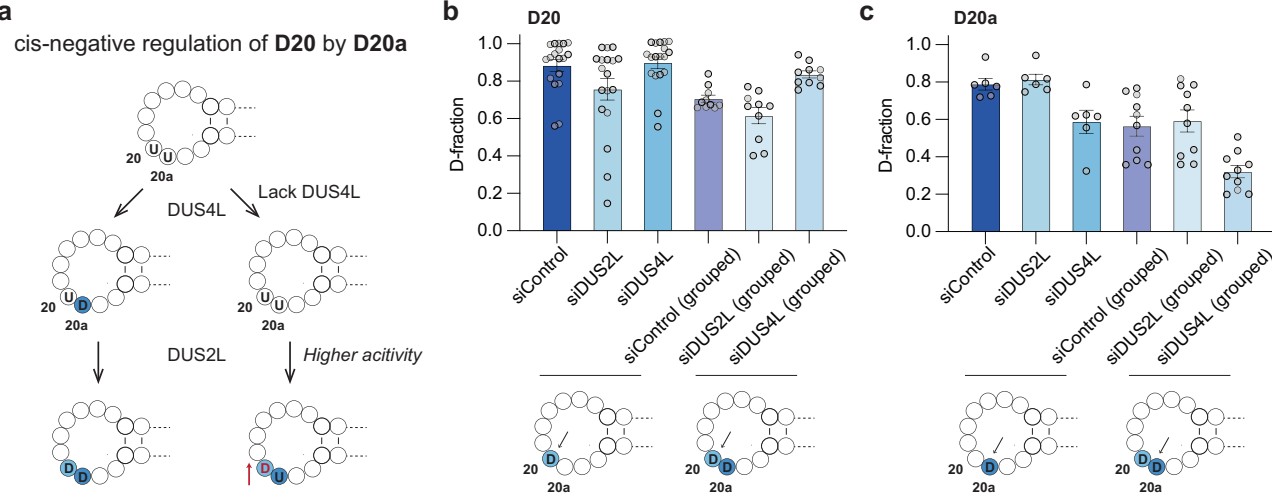

**Fig. 4 | Interaction and co-regulation of adjacent D modifications in human ct-tRNA. a** An integrated overview of the cis-regulation of D20 by D20a, indicating that D20a modification is installed prior to D20. **b** Comparison of D modification levels at D20 sites in HepG2 tRNAs categorized into two groups: "grouped" (tRNAs containing both D20 and D20a sites, $n = 10$) and "others" ($n = 18$), following DUS2L and DUS4L depletion. **c** Comparison of D modification levels at D20a sites in HepG2 tRNAs categorized into two groups: 'grouped' (tRNAs containing both D20 and D20a sites, $n = 10$) and "others" ($n = 6$), following DUS2L and DUS4L depletion. Each point was calculated as the average of at least two biological replicates.

HepG2 small RNA in siControl and siDUS conditions revealed cis-negative regulation of D20 by its neighboring D20a, with no reciprocal regulation observed (Fig. 4a). Since DUS2L and DUS4L were identified as the 'writer' proteins for D20 and D20a modifications, respectively (Fig. 3c, e), HepG2 cytoplasmic tRNAs were categorized into two groups: one with D20 only and the other with both D20 and D20a. For tRNAs containing both D20 and D20a, the reduction of D20a due to the loss of *DUS4L* resulted in increased D modification at D20. This suggests that unmodified uridine at position 20a enhances DUS2L recruitment for D20 deposition (Fig. 4a, b and Supplementary Fig. 7a)[44,45,50]. Pre-tRNA transcripts containing unmodified U20a exhibited a progressive increase in D20 levels during tRNA maturation, whereas transcripts with pre-existing D20a showed minimal changes (Supplementary Fig. 7b–d). These findings, revealed by quantitative CRACI, suggest that D20a can be installed prior to D20 and acts as a cis-negative regulator of D20 formation, but not vice versa (Fig. 4c). This inhibitory effect may stem from D modification-induced changes in local RNA structure, which could affect DUS enzyme recruitment. However, we cannot exclude the possibility that differences in enzymatic activity among DUS variants may also contribute[51].

Interestingly, when we compared tRNA levels by group, we found that tRNAs containing both D20 and D20a modifications are significantly downregulated upon *DUS4L* knockdown (Supplementary Fig. 7e). In contrast, no statistical difference was observed for the expression level of tRNAs with both D20 and 20a modifications in the absence of DUS2L (Supplementary Fig. 7e). Unlike its paralogues and fungal orthologues, DUS2L has acquired an additional domain—a double-stranded RNA binding domain (dsRBD)—which acts as the primary tRNA binding module[52,53]. Published work has shown that this dsRBD is closely associated with the local RNA structure[44,54]. Our findings further suggest that DUS2L's RNA binding function plays a regulatory role beyond its enzymatic activities.

## CRACI compares D quantitative profiles in mammalian species
To study D modifications in different mammalian species, we extracted cellular small RNA (<200 nt) from mES cells and performed CRACI for D detection. The results showed that D modifications are specifically located at positions 16, 17, 20, 20a, 20b, and 47 in mESC cytoplasmic tRNAs, with no D signatures found at other tRNA positions (Fig. 5a and Supplementary Data 4). Analyzing 56 different cytoplasmic tRNAs from mESCs provided a detailed map of tRNA D modifications, including quantification information for each modified site (Fig. 5b and Supplementary Fig. 8a, b). Overall, the distribution pattern of D stoichiometry at these positions in mESC is similar to that observed in HepG2 cells (Fig. 2c). In contrast to the D modification pattern in HepG2 mitochondrial tRNAs, only position 20 in mESC mt-tRNA showed D modification, with two highly modified D20 sites identified in mt-tRNA[Gln] and mt-tRNA[Leu (UUR)] (Fig. 5c, d). When comparing HepG2 cells and mESCs, D16, D17, and D47 exhibited good overlap across mammalian species based on tRNA species and codon usage (Supplementary Fig. 8c, d), while D20a showed weaker overall overlap (Supplementary Fig. 8e). Notably, the tRNA species with D20 modifications in HepG2 cells appeared to be a subset of those with D20 modifications found in mESCs (Supplementary Fig. 8f).

## CRACI reveals D modifications in cytoplasmic, mitochondrial, and chloroplast tRNA in plants
To explore the characteristics of D modifications in mammals and plants, we extracted cellular small RNA and mRNA from *Arabidopsis thaliana* seedlings for CRACI analysis. The cytoplasmic tRNAs of *Arabidopsis thaliana* exhibited D distribution patterns similar to those observed in HepG2 cells and mESCs, primarily at positions 16, 17, 20, 20a, 20b, and 47 (Fig. 5e, Supplementary Fig. 9a and Supplementary Data 5). In contrast to HepG2 cells and mESCs, the mitochondrial tRNAs of *Arabidopsis thaliana* displayed a greater number of D modifications in the D-loop, with observed in more tRNA species and a wider variety of modification sites in the D-loop, specifically at positions 16, 17, 20, and 20a (Fig. 5f, Supplementary Fig. 9b and Supplementary Data 6). Notably, CRACI also detected abundant D modifications at positions 16 and 17 within the D-loop of chloroplast tRNAs in *Arabidopsis thaliana* (Fig. 5g, Supplementary Fig. 9b and Supplementary Data 6). While D modifications were not found in human or mouse rRNA, such as 5S, 18S, and 28S rRNA, CRACI identified a highly modified D2467 site in the chloroplast 23S rRNA of *Arabidopsis thaliana* (Fig. 5h). This finding aligns with the presence of D modifications in bacterial 23S rRNA (D2449), which is located in domain V and shares the same motif (GADAA). This suggests that chloroplasts have retained this feature through a conserved evolutionary process originating from endosymbiotic cyanobacteria[13,55,56]. Overall, D16, D17, D20, and D47 modifications showed significant overlap among HepG2

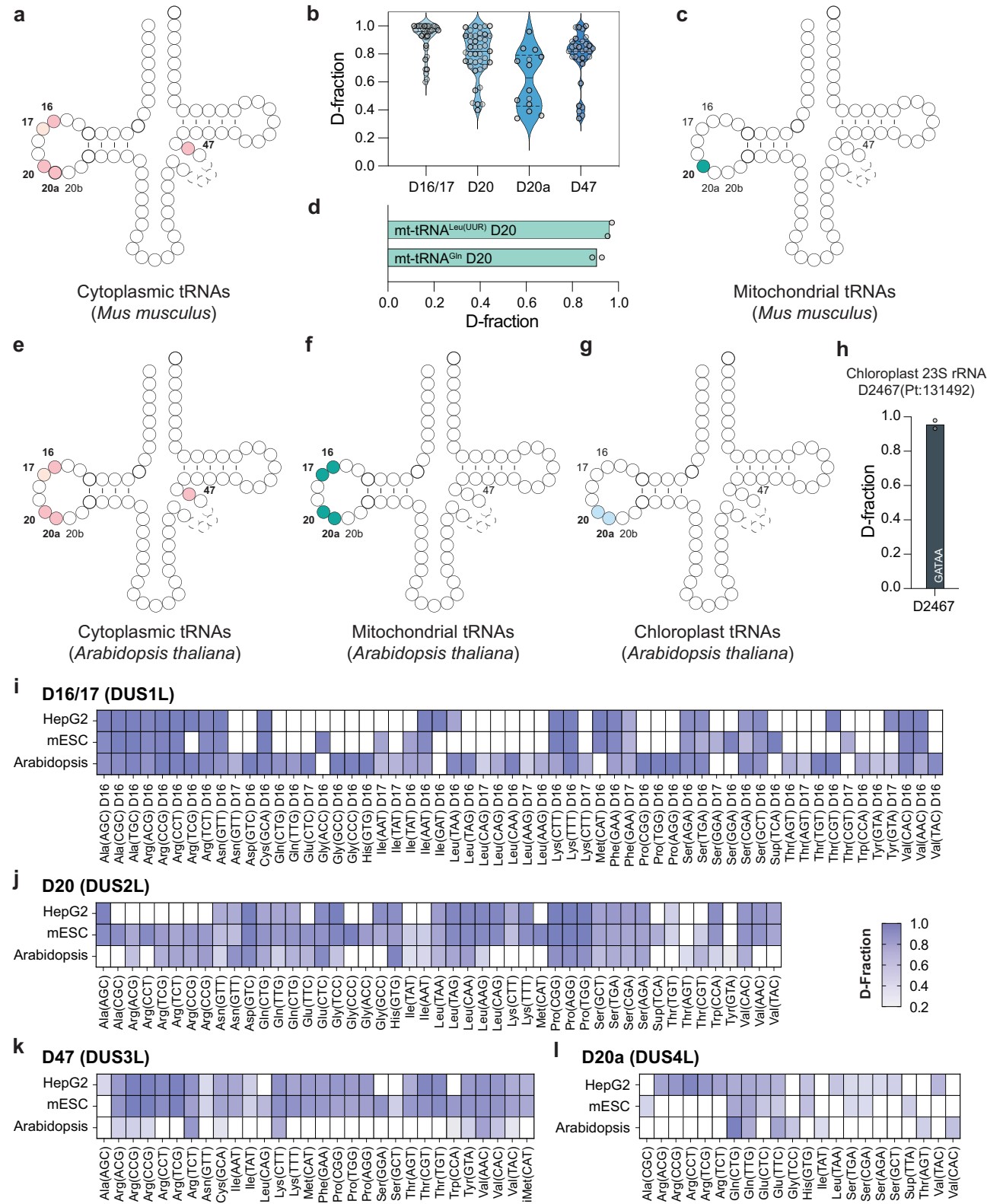

**Fig. 5 | CRACI uncovers D landscape in *Mus musculus* (mESC) and *Arabidopsis thaliana*.** **a** The identified D sites in ct-tRNAs from mouse embryonic stem cells (mESC) using CRACI. **b** Comparison of modification levels at specific D sites in mESC ct-tRNAs, with violin plots representing the distribution of D stoichiometry at different positions of tRNAs. The D fraction of each site was calculated as the average of two biological replicates. **c** The identified D sites in mt-tRNAs from mESCs using CRACI. **d** Quantification of D stoichiometry in mt-tRNAs from mES

cells. The identified D sites in *Arabidopsis thaliana* using CRACI: **e** ct-tRNAs, **f** mt-tRNAs, and **g** chloroplast tRNAs. **h** Detection of D sites in chloroplast rRNA at position U2467. Venn diagrams illustrating the overlap of D sites in ct-tRNAs across 3 species−human (HepG2), mouse (mESC), and plant (*Arabidopsis thaliana*): **i** D16/17, **j** D20, **k** D47, and **l** D20a sites. The D fraction of each site was calculated as the average of two biological replicates.

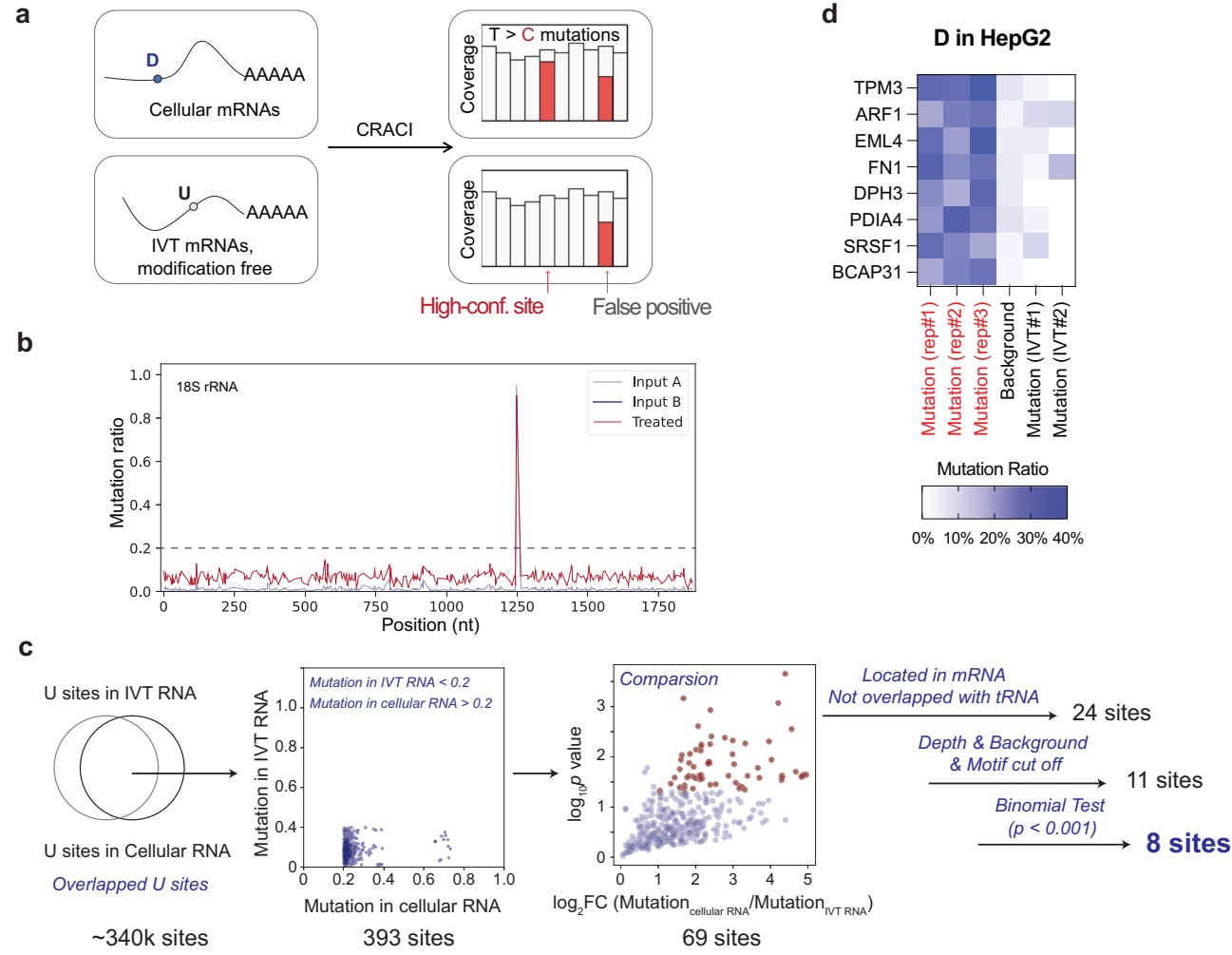

**Fig. 6 | The presence of very rare D modifications in human mRNA. a** Strategic illustration of how in vitro transcribed (IVT) mRNA excludes false positives in D detection by CRACI. **b** Misincorporation signatures revealed by CRACI across 18S rRNA, confirming the absence of false-positive D sites in long RNAs. **c** CRACI analysis pipeline for D detection in HepG2 mRNA. Uridine sites displaying adequate reads coverage in both cellular mRNA and IVT mRNA are considered. We require the misincorporation ratio in IVT mRNA to be less than 0.2 and in cellular mRNA to be greater than 0.2. Differential misincorporation ratios are analyzed, with a log$_2$ fold change >1 and a $p$-value < 0.05 (obtained by two-sided t-test) as cutoffs. Additionally, sites must be in mRNA (not tRNA) and undergo multiple cut-off filtering and binomial testing. **d** Heatmap illustrating D site identified in HepG2 mRNA by CRACI.

cells, mESCs, and *Arabidopsis thaliana*. Plant cells exhibited a more diverse range of D16 and D17 modifications compared to mammals, with the observation of D in tRNA-Asn, tRNA-Gln, tRNA-Glu, tRNA-Gly, tRNA-Leu, tRNA-Pro, tRNA-Thr, tRNA-Tyr, while mammalian cells displayed greater diversity in D47 modifications than plants, which lack D sites in tRNA-Ala, tRNA-Asn, tRNA-Ile, tRNA-Met/iMet, tRNA-Pro, tRNA-Ser, tRNA-Thr (Fig. 5i–k). However, D20a modifications showed a relatively weak overlap between mammals and plants (Fig. 5l).

Although D20b sites have been reported in *S. cerevisiae* Leu and Tyr tRNAs[11], we did not detect any D20b in cytoplasmic tRNAs of *Mus musculus*, *Homo sapiens*, or *Arabidopsis thaliana*. Tyr tRNAs in these species lack a 20b position in their sequences (Supplementary Fig. 10a–d)[57,58]. While certain Leu tRNAs in *M. musculus* and *H. sapiens* do contain a 20b position, CRACI data confirmed the absence of D at these sites (Supplementary Fig. 10e). In *Arabidopsis thaliana*, a mutation signal was observed at Leu 20b in untreated input samples, indicating the presence of other uridine modifications, but not D.

### CRACI identifies rare D modifications in human mRNA

After obtaining comprehensive results from CRACI mapping of D modifications in human and plant tRNAs, we shifted to unraveling the D profiles in mammalian mRNA. To ensure the accuracy of mapping mRNA D modifications and to avoid background mutations in the HepG2 transcriptome, we utilized in vitro transcribed HepG2 polyA$^+$ RNA, devoid of any RNA modifications, as a control for CRACI, referred to as 'IVT'[59]. We also prepared 'input' and 'KBH$_4$ treated' CRACI libraries to uncover D sites in HepG2 mRNA and lncRNA (Fig. 6a). Transitioning from tRNAs (~70 nt) to longer RNAs, the established analysis criteria for D detection in CRACI revealed no D modifications in human rRNAs or other non-coding RNAs like snRNA and snoRNA (Fig. 6b, Supplementary Fig. 10f and Supplementary Data 7). However, the CRACI pipeline identified eight confident D sites within HepG2 mRNA, showing significantly higher misincorporations compared to both 'input' and 'IVT' samples, with D stoichiometry ranging from 10% to 40% (Fig. 6c, d). Addressing the longstanding inquiry into the existence and abundance of D modifications in human mRNA, CRACI provided quantitative evidence at base resolution, consistent with previous mass spectrometry observations[14]. Given that D is typically installed within structured regions of tRNA, we speculate that some D deposition in mRNA may similarly be associated with local secondary structures. However, the observed low modification fraction at these mRNA D sites suggests that they may also arise through non-enzymatic processes, such as the salvage and incorporation of pre-modified D nucleotides derived from tRNA degradation. It will be interesting

to uncover D maps using CRACI in different human and mouse tissue or cell types and examine their functions in the future, with emphasis on mRNA.

## Discussion

In this study, we developed Chemical Reduction Assisted Cytosine Incorporation sequencing (CRACI), a quantitative approach for whole-transcriptome mapping of D modifications at single-base resolution. To the best of our knowledge, CRACI provides the first comprehensive landscape of D modifications in both cytoplasmic and mitochondrial tRNAs from mammals and plants, and enables the identification of specific 'writer' proteins responsible for site-specific D modifications in human cells. Notably, we uncovered D sites in mitochondrial tRNAs and identified DUS2L as the 'writer' enzyme mediating D modifications inside human mitochondria. RNA-seq analysis indicates that D modifications at positions 16, 17, 20, and 47 generally exert minimal effects on tRNA stability, with the exception of the D20a site. Moreover, cross-species comparisons reveal that certain tRNA D modification sites are conserved among humans, mice, and plants.

An intriguing finding from our study is the cis-regulation of D20 by D20a, highlighting a unique instance of crosstalk between the same RNA modification at adjacent sites. This observation underscores the interesting properties of D imparted by its chemical structure, and suggest the importance of DUS2L's RNA binding function as a regulatory role beyond its enzymatic activity.

Furthermore, by leveraging our quantitative CRACI technology alongside IVT RNAs, we addressed the question of whether D modifications exist in mRNAs of human cell lines. Our findings indicated the absence of highly modified D sites within mRNA. The limited number of observed D sites in human mRNA supports the conclusion that D modifications are indeed present, but only at extremely low levels and in restricted locations. However, further studies using primary cells, real tissue samples, and cells under various stress conditions will be necessary to gain a more comprehensive understanding.

Given that D is one of the most abundant and evolutionarily conserved RNA modifications across species, and that DUS enzymes have been implicated in various human diseases, our new method offers a powerful tool to investigate the functional roles of D in gene regulation. Although our CRACI analysis suggests that D has only a modest impact on tRNA stability, its potential involvement in other regulatory processes remains to be explored. Our ongoing studies aim to uncover new functional roles for this ancient uridine modification in diverse biological systems. This work not only offers a comprehensive perspective on D modifications but also establishes a robust sequencing approach as a foundation for future investigations into D biology.

## Methods

### Cell culture

Human HepG2 cells (HB-8065) and mouse embryonic stem cells (mESCs, CRL-1821) were obtained from the American Type Culture Collection (ATCC). HepG2 cells were propagated in DMEM (GIBCO, 11995) containing 10% fetal bovine serum (FBS) and 1% Penicillin–Streptomycin (100×; GIBCO). mESCs were maintained under feeder-free conditions in DMEM (GIBCO, 11995) supplemented with 15% FBS, 1% Penicillin–Streptomycin (GIBCO), GlutaMAX (1×; GIBCO), non-essential amino acids (1×; GIBCO), 2-mercaptoethanol (1×; GIBCO), and 1000 U/mL leukemia inhibitory factor (Millipore, ESG1107). The medium also contained two small-molecule inhibitors: CHIR99021 (3 µM; STEMCELL Technologies; prepared in DMSO) and PD0325901 (1 µM; STEMCELL Technologies; prepared in DMSO). mESCs were grown on 0.2% gelatin-coated plates and subcultured every two days. All cell cultures were maintained at 37 °C in a humidified incubator with 5% $CO_2$.

### Plant materials and growth conditions

Arabidopsis Columbia Col (Col-4) was used in this study. All seeds were sterilized in 10% sodium hypochlorite solution and washed five times with deionized water, then grown on Murashige and Skoog (MS) medium (Cayman Chemical, catalogue no. 16675) supplemented with 0.8% agar and 1.5% sucrose. Seedlings used for the experiments in this study were grown in growth chamber (gBrite™ LED Plant Growth Chambers, 7311-50-2 and Percival, LED-30L1) at 22 °C long day conditions (16 h light / 8 h dark). The 10-day-old seedlings were used to extract RNA.

### DUS siRNA knockdown

The siRNAs for DUS gene knockdown were purchased from Qiagen. HepG2 cells were transfected using siRNAs purchased from Qiagen targeting *DUS1*, *DUS2*, *DUS3*, *DUS4* or non-targeting sequences (Qiagen, AllStars Neg. Control, SI03650318). To prepare the siRNA/RNAi-MAX solution for a 10 cm plate, 40 pmol of siRNAs were diluted in 0.7 mL of OPTI-MEM and 20 µL of RNAiMAX (Thermo Fisher) was diluted in 0.7 mL of the same media in a separate tube. The siRNA and RNAiMAX were mixed together and incubated at room temperature for 15 min. The resulting 1.4 mL transfection solution was added into the HepG2 cell culture which had been cultured for 12-16 hours. RNA was extracted for further analysis 72 hrs after the transfection.

### qRT-PCR

Primers for quantitative reverse transcription PCR (qRT-PCR) were designed to span exon–exon junctions present in all isoforms of the target mature mRNAs. Approximately 200 ng of total RNA was reverse transcribed into cDNA using the PrimeScript RT reagent kit (Takara). The resulting cDNA was then amplified by qPCR with FastStart SYBR Green Master Mix (Roche) on a LightCycler 96 system (Roche). GAPDH served as the internal reference gene. Primer sequences used in this work are provided in Supplementary Data 8.

### RNA isolation

Generally, with harvesting cells as the first step, the media was aspirated, and the cells were washed once with proper volume of ice cold DPBS buffer for each plate. The 10-day-old seedlings and cells were used to extract total RNA with TRIzol reagent (Invitrogen) and then extracted following manufacturer's protocol by isopropanol precipitation. Small RNA fraction was purified from total RNA with RNA Clean & Concentrator kit (Zymo Research) following manufacturer's protocol. mRNA was extracted by two rounds of polyA+ purification with Dynabeads mRNA DIRECT kit (Ambion).

### RNA oligonucleotides preparation

The 35mer-NNDNN RNA probes used in CRACI library preparation for calibration curves were ordered from IDT, with RNase free HPLC purification. Sequence of the probe can be found in Supplementary Data 8.

### D modification fraction estimation

Synthetic 35-mer RNA probes containing the motif –NNDNN– were used as the "100% D" reference, while probes with –NNUNN– served as the "0% D" control. These two standards were combined in defined proportions to prepare six oligonucleotide mixtures representing 100%, 80%, 60%, 40%, 20%, and 0% D content. All mixtures were processed in parallel using the CRACI method. For each sequence context, the mutation frequency was determined, and the relationship between the measured mutation ratio and the D content was fitted to a linear equation: $y = Ax + b$, where y is the observed mutation ratio, x is the D fraction, and A and b are fitted parameters. Values for A and b corresponding to each sequence context are provided in Supplementary Data 1.

## CRACI for D site detection

Wild-type HepG2, mESC, and *Arabidopsis thaliana* seedling cells were prepared in duplicate for each sample type, with one 10-cm plate per replicate. The same setup was used for HepG2 siControl and DUS knockdown cells. From each sample, either small RNAs (<200 nt) were enriched by size selection, or poly(A)+ RNAs were obtained by oligo-dT pulldown. Approximately 100 ng of RNA from mammalian or plant samples was fragmented using RNA Fragmentation Reagents (Invitrogen, AM8740) at 70 °C for 14 min and purified with the Oligo Clean & Concentrator kit (Zymo Research). The 3′ ends were repaired and the 5′ ends phosphorylated using T4 polynucleotide kinase (PNK; Thermo Fisher Scientific, EK0032). For this reaction, RNA was combined with 3 μL of 10× T4 PNK reaction buffer (NEB, B0201S) and 3 μL T4 PNK in a final volume of 30 μL, incubated at 37 °C for 45 min, then supplemented with 1.5 μL T4 PNK and 1.5 μL 10 mM ATP for an additional 45 min at 37 °C, followed by purification with Oligo Clean & Concentrator and elution in 10 μL RNase-free water.

For 3′-adapter ligation, 10 μL of the repaired/phosphorylated RNA was mixed with 1.0 μL of 20 μM RNA 3′ SR Adapter (5′App-NNNNNATCACGAGATCGGAAGAGCACACGTCT-3SpC3; inline barcode ATCACG), heated at 70 °C for 2 min, and placed on ice. The ligation mix contained 2.5 μL 10× T4 RNA Ligase Reaction Buffer (NEB, M0373L), 7.5 μL PEG8000 (50%), 1 μL SUPERase•In RNase inhibitor, and 2 μL T4 RNA Ligase 2 truncated KQ (NEB, M0373L). Reactions were incubated at 25 °C for 2 h and then at 16 °C for 10 h. Excess adapters were removed by adding 2 μL 5′-deadenylase (NEB, M0331S) at 30 °C for 45 min, followed by 1 μL RecJf exonuclease (NEB, M0264L) at 37 °C for 45 min. RNA was again purified with RNA Clean & Concentrator and eluted in 10 μL RNase-free water. The purified RNA was incubated with 1.2 μl 10 μM 5′ SR Adapter (5′-GUUCAGAGUUCUACAGUCCGACGAUC-3′) at 70 °C for 2 mins and placed immediately on ice. Then 2.5 μl 10× T4 RNA ligase reaction buffer, 1.0 μl 25 mM ATP, 10 μl PEG8000 (50%) and 1 μl T4 RNA Ligase 1 (high concentration, catalog no. M0437M, NEB) were added to the RNA–adapter mixture. The reaction was mixed well and incubated at 25 °C for 8 h, followed by RNA Clean and Concentrator (Zymo Research) purification, eluting with 12 μl RNase-free water.

From the purified RNA, 2 μL was allocated for "Input1" libraries, 2 μL for "Input2" libraries, and the remaining 8 μL underwent CRACI-optimized reduction. For this treatment, RNA was combined with 40 μL freshly prepared 1 M KBH₄ (54 mg KBH₄ dissolved in 1 mL RNase-free water, pH -7.5) and incubated at 25 °C for 3 h. RNA was then purified and eluted in 10 μL water. "Input1," "Input2," and treated samples were diluted to 10 μL, mixed with 1.0 μL 2.0 μM SR RT primer (5′-AGACGTGTGCTCTTCCGATCT-3′), heated at 65 °C for 2 min, and placed on ice. For "Input2" and treated samples, reverse transcription was performed in a mix containing 2 μL 10× AMV RT Buffer (NEB, B0277AVIAL), 2 μL 1 mM dNTP mix (from NEB N0447L), 2 μL 10 mM dGTP (from NEB N0442S), 1 μL RNaseOUT (Thermo Scientific, 10777019), and 2 μL HIV RT (Worthington LS05003). For "Input1" libraries, the reaction contained 2 μL RNase-free water, 2 μL 10× AMV RT Buffer, 2 μL 10 mM dNTP mix, 1 μL RNaseOUT, and 2 μL HIV RT.

The reaction was incubated at 70 °C for 5 min, after which cDNA was purified using the DNA Clean & Concentrator kit (Zymo Research). Purified cDNA was eluted in 20 μL and stored at −80 °C. For library preparation, 4 μL of cDNA was used per 15-cycle PCR reaction with the SR Primer for Illumina (NEB) and indexed primers from the NEBNext Multiplex Oligos for Illumina kit. Final libraries were size-selected on 3.5% low−melting point agarose gels and sequenced on an Illumina NovaSeq X platform using single-end 100 bp reads.

## Modification-free mRNA fragments by in vitro transcription (IVT)

The polyA+ RNA was fragmented using RNA Fragmentation Reagents (Thermo Fisher Scientific) at 95 °C for 3.5 min and cleaned up by RNA Clean & Concentrator column (Zymo Research). Then RNA fragments were end-repaired using PNK and then ligated with a 3′- adaptor. The 5′-adaptor containing T7 promoter sequence was then ligated to the 5′-end of RNA fragments. Next, the first and the second-strand cDNA was synthesized using SuperScript IV and Q5 DNA polymerase, respectively. Double-stranded cDNA was purified using the DNA Clean & Concentrator (Zymo Research). The T7 transcription step was carried out using T7 RNA polymerase (NEB) at 37 °C for 1 hour. The dsDNA in the reaction mixture was then cleaned up using TURBO DNase (Thermo Fisher Scientific) and the resulting IVT RNA was purified by RNA Clean & Concentrator (Zymo Research). The IVT RNA was then used for CRACI library preparation.

## Mitochondrial tRNA isolation and enrichment

Mitochondria were prepared from four 15-cm culture dishes of HepG2 cells using the Mitochondria Isolation Kit for Cultured Cells (Thermo Fisher Scientific, 89874). RNA was then extracted from the mitochondrial fraction with TRIzol reagent (Invitrogen) according to the manufacturer's instructions, followed by isopropanol precipitation. The small RNA fraction was subsequently enriched from total RNA using the RNA Clean & Concentrator kit (Zymo Research) in accordance with the supplier's protocol.

Biotinylated single-stranded DNA probes are synthesized from IDT with the sequence from published paper (Supplementary Data 8)[49]. The ASO-enrichment protocol was adapted from previously published methods[60,61]. Briefly, 20 μl of RNase-free Dynabeads (Invitrogen) were prepared, washed with buffer A (10 mM Tris-HCl, pH 7.5, 2 mM EDTA, 2 M NaCl) and finally resuspended in 20 μl of buffer A. Biotinylated antisense oligonucleotides (200 μM in 10 μl water) were mixed with an equal volume of Dynabeads and incubated at room temperature for 30 min. Then the oligonucleotide-coated beads were then washed with buffer B (5 mM Tris-HCl, pH 7.5, 1 mM EDTA, 1 M NaCl) and equilibrated in 6× SSC. The oligonucleotide-coated beads and isolated mitochondria small RNA in 6× SSC were heated separately to 75 °C for 10 min, then combined and incubated at 75 °C for another 10 min, followed by 3 h at room temperature. After hybridization, beads were washed sequentially with 3×, 1×, and 0.1× SSC until the wash was UV-clear. Bound tRNAs were eluted twice with 10 μl RNase-free water.

## Quantitative analysis of RNA modifications by LC-MS/MS

Enriched mitochondrial tRNAs were digested with nuclease P1 (MilliporeSigma, N8630) in 20 μl of 20 mM ammonium acetate buffer (pH 5.3) at 42 °C for 2 h. Subsequently, 1 U of FastAP thermosensitive alkaline phosphatase (Thermo Fisher Scientific, EF0651) and 1× FastAP buffer were added, followed by incubation at 37 °C for 2 h.

LC-MS/MS analysis was performed with Agilent 1290 ultrahigh pressure liquid chromatography system coupled to an Agilent 6495 triple quadrupole mass spectrometer or Agilent 6460 LC-MS/MS spectrometer. Chromatographic separation was achieved by using a Waters Atlantis T3 column (2.1 ×100 mm, 1.7 μm particle size) with 0.1% formic acid in water as the mobile phase at a flow rate of 0.3 mL/min. The column was kept at 40 °C and the auto-sampler was cooled at 4 °C. The electrospray ionization of the mass spectrometer was performed in positive ion mode with the following source parameters: drying gas temperature 200 °C with a flow of 14 L/min, nebulizer gas pressure 30 psi, sheath gas temperature 400 °C with a flow of 11 L/min, capillary voltage 3,000 V and nozzle voltage 500 V. Compounds were detected in multiple reaction monitoring (MRM) mode with the following transitions: m/z 247.1 to 115.1, 247.1 to 97 for D and m/z 249.1 to 117.1, 249.1 to 56.1 for THU. Data acquisition and processing were performed using MassHunter software (Agilent Technologies).

## Sequencing data processing and analysis for ncRNAs

All sequencing reads were first processed with Cutadapt (v4.8) to remove adapter sequences and low-quality bases. PCR duplicates were

eliminated using BBMap (v38.73). Five-nucleotide random barcodes at the read ends were then trimmed, and reads shorter than 20 nt or of low quality were discarded with Cutadapt. For tRNAs, remaining reads were aligned to tRNA sequence obtained from GtRNAdb (https://gtrnadb.ucsc.edu/) using HISAT-3N (v.2.2.1-3n-0.0.3). Mitochondria and chloroplast tRNA sequences are obtained from published papers. Meanwhile, we have also aligned remaining reads to hg38, mm10, or TAIR10 genome using HISAT-3N to investigate D sites in other RNA species. The generated table from HISAT-3N were parsed and analyzed by inhouse scripts. Internal mutation ratio at each D candidate site suggested by HISAT-3N, was further confirmed by direct IGV visualization (v.2.8.0).

In summary, in tRNA or other ncRNA, one D candidate site needs to satisfy the following criteria in its mutation profile: (1) mutation ratio above 20% (with misincorporation count above five) in CRACI libraries; (2) mutation ratio below 15% in 'Input1' and 'Input2' libraries; (3) total reads coverage depth above 50 in both 'Treated' and 'Input1'/'Input2' libraries; (4) mutation ratio in 'Treated' libraries must be above 3-fold over background in any given sequence motif (defined as the misincorporation rates detected from RNA probes containing 0% D as in Supplementary Fig. 1c); (5) mutation ratio in 'Treated' libraries should be 3-fold over mutation ratio in 'Input2' libraries; (6) we excluded uridine sites with multiple U motif due to the mutation will be accumulated during analysis; (7) all mutation ratio must be from 'U' sites marked instead of from A or C or G. The 'input1' samples of CRACI are equivalent to regular RNA-seq; therefore, we quantified the gene-level read counts of input samples that aligned to genome for gene expression analysis with DESeq2.

### Sequencing data processing and analysis for mRNAs with IVT as background

All sequencing reads were first processed with Cutadapt (v4.8) to remove adapter sequences and low-quality bases. PCR duplicates were eliminated using BBMap (v38.73). Five-nucleotide random barcodes at the read ends were then trimmed, and reads shorter than 20 nt or of low quality were discarded with Cutadapt. Then the remaining reads have been aligned to hg38, mm10, or TAIR10 genome using HISAT-3N. The generated table from HISAT-3N were parsed and analyzed by inhouse scripts. Internal mutation ratio at each D candidate site suggested by HISAT-3N, was further confirmed by direct IGV visualization (v.2.8.0).

To identify D sites in mRNA, we required the 'U' sites are existed in both IVT RNA and cellular RNA samples. Then the D candidate site needs to satisfy the following criteria in its mutation profile: (1) mutation ratio above 20% in CRACI libraries of cellular RNAs; (2) mutation ratio below 20% in CRACI libraries of IVT RNAs; (3) The p value between mutation ratio in cellular RNA and mutation ratio in IVT RNA should be below 0.05 and the mutation ratio in cellular RNA should be above 3-fold over mutation ratio in IVT RNA; (4) The potential D sites must not be overlapped with tRNA regions in the genome; (5) Mutation ratio in cellular 'Treated' libraries must be above 2-fold over background in any given sequence motif and 'Input2' libraries; (6) total reads coverage depth above 30 in cellular 'Treated' libraries; (7) we excluded potential sites with multiple U motif due to the mutation will be accumulated during analysis; (8) Binomial test is conducted between the depth, mutation ratio, and the background.

### Statistics and reproducibility

For CRACI libraries, two or three biologically independent replicates were used in each experiment with cultured cells. Data are presented as the mean ± s.d., with two-tailed Student's t-tests on the statistical significance of differences between groups unless specific note. Paired t-tests are used for comparing D ratio at same sites upon DUS KD. All statistical analysis and data graphing were done in Prism (v.9.2.0) software.

No statistical methods were applied to pre-evaluate sample size. No data were excluded from analysis. Samples in this study were not randomized. Blinding was not used for this study because cell culture, sample preparation, reagents and experimental settings were kept consistent for each experiment.

### Reporting summary

Further information on research design is available in the Nature Portfolio Reporting Summary linked to this article.

## Data availability

The raw and processed sequencing data have been deposited into the NCBI Gene Expression Omnibus (GEO) database with the accession number GSE278487. Source data are provided with this paper.

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

## Acknowledgements

We thank Dr. Pieter W. Faber, Dr. Lindsay Scarpitta, and the staff at the Genomics Facility and Comprehensive Cancer Center sequencing facility at the University of Chicago for NGS sequencing measurement. C.H. thanks the Ludwig Center for Metastasis at the University of Chicago and Rolfe Foundation for their support. We thank support from the Research Grants Council (RGC) of Hong Kong for grant ECS 26103623 (L.-S.Z.). We thank support from the National Institute of Health (NIH) grants RM1 HG008935 (C.H.). C.H. is an investigator at Howard Hughes Medical Institute.

## Author contributions

C.H., C.-W.J. and L.-S.Z. conceived the project. C.-W.J., L.-S.Z. and H.L. developed the experimental protocol. C.W.J. and L.-S.Z. developed the analysis methods. C.W.J. and H.L. constructed the sequencing libraries. B.J., X.Z., L.C., Z.H., J.Z., Y.L., S.S., H.S., C.Y., Y.Z., R.G., P.X., Y.J., S.L., F.Y., B.L., Y.X. and J.W. contributed to the experiments and analyzed the results. C.-W.J., L.-S.Z. and C.H. wrote the manuscript with input from all authors.

## Competing interests

C.H. is a scientific founder, a member of the scientific advisory board and equity holder of Aferna Bio, Inc., AllyRNA, Inc., and Ellis Bio, Inc., a scientific cofounder and equity holder of Accent Therapeutics, Inc., and a member of the scientific advisory board of Rona Therapeutics. The remaining authors declare no competing interests.
