## [Transparent Peer Review file · Nature Communications]

Quantitative CRACI reveals transcriptome-wide distribution of RNA dihydrouridine at base resolution

Corresponding Author: Professor Chuan He

Version 0:

Reviewer comments:

Reviewer #1

(Remarks to the Author)

The manuscript by Ju et al., presented an optimized method for RNA dihydrouridine (D) sequencing called CRACI. Unlike previous methods that generate a truncation signal at D, CRACI generates a D-to-C mutation signal to achieve base-resolution and quantitative sequencing of D. The authors showed that CRACI has very high conversion on D, relatively low false positives, and excellent linear calibration curves for accurate quantification of D. The authors then applied it to human, mouse, plant RNA, mainly tRNA, to identify novel D sites and investigated the writer proteins (DUS enzymes). Overall, this is an exciting method for D sequencing which could aid in future studies of D modifications in RNA. I have a few comments.

1. The authors used a known KBH₄ reduction chemistry and combined it with a new RT enzyme (HIV RT) and a high dGTP/dNTP ratio to achieve a high D-to-C conversion. Could the authors comment on why reduced D would prefer to pair with G, and which factor contributed most to the high mutation rate?
2. HIV RT is not a high-fidelity enzyme and the high dGTP/dNTP ratio will further increase the error rate. This high background of additional mutations may affect certain applications. Could the authors provide a breakdown of the mutation rates (including stop signal) at D and other canonical bases? For example, the authors could use rRNA or synthetic RNA and plot the mutation rates at ALL nucleotides, similar to Figure 6b.
3. The authors suggested D20a could cis-negatively regulate D20. This implies D20a is generated before D20. Could the authors look into their data for evidence supporting this order? Perhaps by analyzing the reads mapped to pre-tRNA.
4. Some figure references in the main text are incorrect. Line 174, Fig 2h should be Fig 2i. Line 179, Fig 2i,j should be Fig 2j,k.

Reviewer #2

(Remarks to the Author)

The article by Cheng-Wei Ju and collaborators introduces the development and application of an innovative method named "Chemical Reduction Assisted Cytosine Incorporation sequencing" (CRACI). This approach aims to quantitatively and comprehensively map dihydrouridylation sites within the transcriptome. The methodology relies on a chemical pre-reduction of the dihydrouracil (D) base in a given RNA molecule to tetrahydrouridine using potassium borohydride (KBH₄). In the presence of high GTP concentrations, this reduction enables HIV reverse transcriptase to incorporate a guanine (G) opposite tetrahydrouridine, leading to a detectable misincorporation during high-throughput sequencing.

This technique was applied to total RNA from human HepG2 cells, mouse embryonic stem cells (mESCs), and the model plant *Arabidopsis thaliana* to establish a detailed mapping of dihydrouridylation sites in cytosolic, mitochondrial, and chloroplast RNAs. The results confirmed expected observations based on previous tRNA sequencing data, such as: (i) The presence of D in cytosolic tRNAs at canonical dihydrouridylation positions (16, 17, 20, 20a, and 47); (ii) The localization of D20 in mammalian mitochondrial tRNAs. In parallel, the study unveiled novel and quite interesting equally important findings on the metabolism of dihydrouracil in tRNAs: (i) The presence of D at non-canonical positions (16 and 17) in mitochondrial tRNAs from HepG2 cells. (ii) A negative dependency between D20a and D20, suggesting an

interdependence between these modifications. (iii) The coexistence of D16, D17, D20, and D20a in mitochondrial tRNAs of *A. thaliana* and D20 and D20a in chloroplast tRNAs. Using siRNA targeting the Dus enzymes (dihydrouridine synthases), the authors identified site-specificities for these enzymes. Most of the observations were consistent with known data, except for Dus2, which demonstrated a remarkable expanded specificity in a specific cellular context (HepG2 cells). In addition to its known role in depositing D20, Dus2 was shown to introduce D16 and D17, a plasticity not observed in mESCs. Beyond tRNAs, CRACI enabled exploration of dihydrouridylation in mRNAs, a controversial issue, confirming the presence of D in these RNAs, albeit at very low levels and in a limited number of transcripts. However, this technique does not detect dihydrouridylation sites in other types of non-coding RNAs, unlike other methodologies such as D-seq. Finally, a particularly innovative finding is the identification of D2467 in the 23S chloroplast rRNA of *A. thaliana*, marking the first report of this modification in eukaryotes, although it has been described in bacterial rRNAs.

These results represent a significant advancement in the study of RNA dihydrouridylation. CRACI stands out from existing methods due to its specificity, quantitative nature, and the absence of abasic sites induced during the process. The conclusions are compelling and mark a major progress in a field that remains underexplored, despite the abundance of the D base in the transcriptome, second only to pseudouridine. This method opens promising new avenues for investigating dihydrouridylation metabolism under physiological and pathological conditions. Consequently, I believe this article deserves publication in *Nature Communications* given the importance of the results obtained.

Important comments for improvement:

Absence of D20b: The authors did not detect D20b in cytosolic tRNAs, although its existence has been reported. Do they have an explanation? This could reveal a limitation of the method, particularly for certain RNA types or specific sites, and should be mentioned in the revised manuscript.

KBH4 treatment conditions: The reduction of dihydrouridine to tetrahydrouridine depends on pH. The authors should specify the pH conditions used, as under basic pH, hydrolysis of the dihydrouracil heterocycle could produce ureidopropional compound, which undergoes β -elimination, resulting in an abasic site.

Orthogonal validation: While the results are robust, it would be valuable to confirm some findings using an independent method, such as MALDI-MS, particularly the presence of D16 and D17 in mitochondrial tRNAs from HepG2 cells. I understand this might be challenging due to the low quantities of mitochondrial tRNAs and the low stoichiometry of D, but it would be worth attempting.

Integration of recent work: The authors should reference the recent study on hDus1L (Matsuura J et al., *Commun Biol.* 2024 Oct 2;7(1):1238. doi: 10.1038/s42003-024-06942-8).

Minor points:

Regarding functional redundancy among human Dus enzymes, the genes were not fully deleted and it is hard to judge. Recently, functional redundancy has been seen in Dus from *B. subtilis* but not in *E. coli*. Could the authors comment on this aspect, particularly how siRNA targeting induces compensation in D levels and which sites?

Line 50, DUS enzymes are not synthetases but synthases (dihydrouridine synthetases). Please make the correction. By convention synthetases are enzymes that employ ATP in the catalyzed reaction like for instance amino-acyl tRNA synthetases.

Line 50: Reference 8 discusses kinetic studies on a single Dus enzyme, Dus2p, but does not address the issue of selectivity between NADPH and NADH. While Dus enzymes preferentially use NADPH, they can also utilize NADH, albeit with a lower K_M . This is clarified in Reference 2.

Line 51: Reference 9 is not cited correctly. This work does not discuss DusA, DusB, or DusC. Furthermore, most bacteria do not possess all three Dus enzymes. The presence of all three Dus enzymes is specific to Proteobacteria. In contrast, as noted in Reference 10, Gram-positive bacteria possess only DusB and lack both DusA and DusC.

Line 177: Replace "mass spec" with "mass spectrometry."

Line 285: Correct "DUC" to "DUS."

Reviewer #3

(Remarks to the Author)

This paper co-led by Li-Sheng Zhang and Chuan He describes new Chemical Reduction Assisted Cytosine Incorporation seq approach, abbreviated CRACI, which maps D modifications in the RNAome. This approach utilizes chemical reduction with KBH4, previously applied to N4-acetylcytidine, to reduce D. It is followed by reverse transcriptase reactions that induce mutations at RNA modification sites in the presence of elevated GTP (1 mM) to dNTP (10 μ M) ratios, yielding a reliable sequencing method for identifying D modifications

The D modification is an abundant feature in tRNAs, which are the primary species detected using CRACI. Through siRNA-based silencing of individual DUS enzymes, the authors comprehensively map tRNA modifications of both cytoplasmic and mitochondrial origin in two eukaryotic cell lines, uncovering several new modification sites in mito-tRNAs. They also apply the method to *A. thaliana* seedlings, revealing a similar distribution of D modifications between cytoplasmic tRNAs of mammals and plants, but notable differences in mito-tRNAs. *A. thaliana* mito-tRNAs are more extensively modified than their mammalian counterparts. Ribosomal RNA and ncRNAs lack D modifications, except for chloroplast 23S rRNA in *A. thaliana*, which resembles bacterial 23S D2449. Although a few low-level modifications were detected in human mRNA, their origin remains unclear.

In summary, this is well-written manuscript with a straight forward experimental design and convincing results. However, a few aspects require clarification before considering the manuscript for publication:

1. In the section describing the quantification of the D stoichiometry (from l. 131 on), it is unclear why 5-nt motif has been considered. Some reports (<https://pmc.ncbi.nlm.nih.gov/articles/PMC10635142/>) show that motifs as small as two nucleotides (GU) could be recognized by DUS. The authors should explain their rationale for selecting the motifs and the

filtering scheme.

2. The sequencing protocol used in this study is not fundamentally new, and the original protocols should be cited. The sequencing approach with direct adaptor ligation was originally developed for miRNA and now is widely employed in ribosome profiling (Ribo-seq) protocols. The original publications should be appropriately acknowledged.

3. Since this is a methodological paper, it should provide more explanation on how in a polyA-selection step small RNAs are retained. Are small RNAs quantitatively retained? Why polyA selection is chosen over rRNA depletion? The first part of the paper would definitely benefit from more thorough explanation of the steps supported by evidence on how quantitative each fraction is?

4. In the section 'Quantitative CRACI uncovers cytoplasmic and mitochondrial tRNA' (l. 150-171), the authors detect different number of D modifications in the tRNAs, with some having only a single modification. A correlation with tRNA abundance would be helpful to demonstrate that a lower number of modifications is not associated with low-abundance tRNAs.

Additionally, the authors should compare the D modification map generated by CRACI with previously described D positions in mammalian tRNAs, as documented in Modomics and GtRNAdb.

In the same paragraph, which are the tRNAs outside the 10 top matches, that do not overlap with the CLIP/iCLIP data? A list of those would be helpful.

5. In the paragraph 'CRACI assigns the 'writer' proteins to D modifications in human cytoplasmic and mitochondrial tRNAs' (l. 185-201), the levels of DUS enzymes were silenced by siRNA to different extent. The expression levels of DUS1L and DIS4L were particularly low; however, the stoichiometry of the corresponding D positions remains fairly high. Only a few tRNAs appear sensitive to the absence of these enzymes, while for the majority, the modification levels remain at nearly 100% (Extended 4 and 3b-e). This unexpected result suggests possible cross-reactivity among DUS enzymes. A critical control is missing here: the authors should silence all DUS enzymes together and demonstrate that D modifications are nearly absent (<https://academic.oup.com/nar/article/52/21/12784/7845166>). This would serve as an important control to validate the sensitivity of CRACI.

In the same paragraph, the claim that the study establishes '...DUS2L as the sole active 'writer' protein responsible for D modifications in HepG2 mitochondria' should be softened, as the data do not clearly support this strong claim.

6. The conclusion in l. 220-221 regarding the secondary structure is somewhat overstated, as there is no direct evidence supporting the involvement of these regions in a secondary structure. Alternatively, this could simply reflect differences in enzymatic activities (<https://academic.oup.com/nar/article/52/21/12784/7845166>) - a scenario that the authors should consider in their discussion.

7. The D sites in human mRNA are sparse. Have those 8 sites been found in previous studies? What is the reproducibility of detecting these sites across biological replicates? Additionally, the conclusion regarding potential secondary structures seems too strong and lacks experimental evidence (e.g., structure-seq data). The authors should tone it down and also discuss alternative possibilities. It cannot be ruled out that D incorporation in mRNA is not enzymatic, and incorporated as pre-modified D nucleotide (e.g., originating from degradation of the abundant tRNA^{ome}) during mRNA synthesis.

8. The legend of each figure/panel should provide information on the number of biological replicates. There are many figures without SD. Do they represent merged replicates?

Version 1:

Reviewer comments:

Reviewer #1

(Remarks to the Author)

The authors have addressed my comments in the revision.

Reviewer #2

(Remarks to the Author)

The authors have convincingly addressed all of my concerns, and the paper now merits publication in Nature Communications. However, there are a few minor corrections that should be made before final acceptance:

Figure 1C: There are two labeling errors: "DMTP" should be corrected to "dNTP".

Line 127: What does "pf" stand for? This should be corrected.

Figure 2C: It would have been insightful to also include D17 and D20a for comparison.

Line 271: References 9 and 50 do not support the claimed role of hDus2's dsRBD in tRNA structure recognition. The correct references are 44 and the following study: *Biochemistry* 2019, *58*(20), 2463–2473. DOI: 10.1021/acs.biochem.9b00111.

Reviewer #3

(Remarks to the Author)

With the addition of several new experiments and analyses, the authors have adequately addressed all concerns and comments from the previous round.

Point-by-point Response

We want to thank the reviewers for their constructive comments and critiques, which have helped to significantly improve our current version of the manuscript. We include a summary at the beginning to describe our newly added experiments and analyses performed to address key comments raised by the editor and reviewers. We have updated and added 23 new figure panels in extended data figures and revised the corresponding paragraphs (in blue font) in the manuscript accordingly.

Summary of new experiments:

1. Technical validation of CRACI and background control

- Demonstrated low false-positive rates and absence of RT truncation artifacts using untreated rRNA controls (Extended Data Fig. 2).
- Validated the KBH_4 -mediated chemical reduction by HPLC in model reactions (Extended Data Fig. 1a).
- Performed LC-MS/MS on ASO-enriched mt-tRNAs to validate the newly identified mitochondrial D sites (Extended Data Fig. 4b).

2. Validation with external datasets

- D sites identified by CRACI showed strong overlap with Modomics (27 out of 30); unmatched sites suggest novel or misannotated positions (Extended Data Fig. 3i-3j).
- Provided detailed comparison between CRACI-detected D sites and iCLIP targets of corresponding writer enzymes (Extended Data Fig. 3f-3h).

3. Investigate pre-tRNA dynamics and functional redundancy

- Suggest D20a as a cis-negative regulator of D20 during tRNA maturation through pre-tRNA analysis (Extended Data Fig. 7b-7d).
- Performed combinatorial knockdown of *DUS1L/2L/3L/4L*, resulting in reduced mutation ratio for >70% of D sites (Extended Data Fig. 5f).

4. Expanded methodological clarifications

- Discussed the absence of D20b sites in *Homo sapiens* (Extended Data Fig. 10a-10e).
- Confirmed that D site detection is not biased because of varied tRNA expression levels (Extended Data Fig. 3c)

Response to comments from reviewers

Reviewer #1: *The manuscript by Ju et al., presented an optimized method for RNA dihydrouridine (D) sequencing called CRACI. Unlike previous methods that generate a truncation signal at D, CRACI generates a D-to-C mutation signal to achieve base-resolution and quantitative sequencing of D. The authors showed that CRACI has very high conversion on D, relatively low false positives, and excellent linear calibration curves for accurate quantification of D. The authors then applied it to human, mouse, plant RNA, mainly tRNA, to identify novel D sites and investigated the writer proteins (DUS enzymes). Overall, this is an exciting method for D sequencing which could aid in future studies of D modifications in RNA. I have a few comments.*

Response: We would like to thank the reviewer for the positive comments.

1. *The authors used a known KBH₄ reduction chemistry and combined it with a new RT enzyme (HIV RT) and a high dGTP/dNTP ratio to achieve a high D-to-C conversion. Could the authors comment on why reduced D would prefer to pair with G, and which factor contributed most to the high mutation rate?*

Response: We thank the reviewer for the suggestions. While the canonical base pairing is U–A, U–G wobble base pairing can occur under certain circumstances, although it is disfavored in most cases. Dihydrouridine (D), even in its native form, attenuates the conventional U–A pairing due to its increased conformational flexibility and altered hydrogen bonding. Upon KBH₄-mediated chemical reduction, the reduced D becomes less planar and more structurally distorted, further weakening its pairing with A. Based on our results, this distortion appears to promote mispairing with G instead, leading to an increased frequency of U-to-C conversions during reverse transcription.

While it is challenging to quantify the contribution of each factor, our data suggest that KBH₄ reduction, the use of HIV RT, and high dGTP/dNTP ratio are critical for inducing high D-to-C mutations. The KBH₄ reduction step modifies the base to favor mispairing, while HIV-RT facilitates the incorporation of G opposite the reduced D in the presence of high dGTP/dNTP ratio. These are the main findings leading to GRACI.

We have included discussion on page 5 of the revised manuscript: 'The KBH₄ reduction renders D less planar and destabilizes U–A pairing, shifting it to mispair with G (**Extended Data Fig. 1b**). The permissive readthrough by HIV reverse transcriptase, in the presence of elevated dGTP/dNTP ratio, further enhances T→C misincorporation at the reduced D sites.'

Extended Data Figure 1. Optimization of RT conditions in CRACI.

b. Proposed base-pairing model between guanosine (G) and reduced D

2. HIV RT is not a high-fidelity enzyme and the high dGTP/dNTP ratio will further increase the error rate. This high background of additional mutations may affect certain applications. Could the authors provide a breakdown of the mutation rates (including stop signal) at D and other canonical bases? For example, the authors could use rRNA or synthetic RNA and plot the mutation rates at ALL nucleotides, similar to Figure 6b.

Response: We thank the reviewer for the suggestions. We have included the detailed analysis in the revised manuscript as the follows: 'To evaluate potential false-positives in CRACI, we analyzed mutation profiles across cellular 18S and 28S rRNA in treated samples. No significant increases in canonical base mutation rates (based on the defined cut-offs) nor any notable RT truncation signals were observed (Extended Data Fig. 2).'

Extended Data Figure 2. Evaluation of the background of CRACI using cellular 18S rRNA.

a. Mutation ratio across the 18S rRNA in input and treated samples. Known modified positions such as $m^1acp^3\Psi$ and m^7G are highlighted. The mutation ratio was calculated as the average of three biological replicates. **b.** Normalized read coverage across 18S rRNA for input and treated conditions, suggesting consistent coverage and the absence of major RT truncations. The reads coverage was calculated as

the average of three biological replicates. **c.** Boxplot showing base mutation frequencies (G, A, C, T) in 18S rRNA for each sample. **d.** Mutation ratio at the m⁷G1639 across different conditions. **e.** Mutation ratio at the ac⁴C1337 across different conditions.

3. The authors suggested D20a could cis-negatively regulate D20. This implies D20a is generated before D20. Could the authors look into their data for evidence supporting this order? Perhaps by analyzing the reads mapped to pre-tRNA.

Response: We thank the reviewer for these suggestions. To investigate this temporal order and potential cis-regulatory relationship between D20a and D20, we first analyzed mature tRNA molecules. We confirm that, compared to tRNAs that contain D20a, tRNAs with unmodified U20a consistently exhibited higher D modification levels at position 20. To examine D installation during tRNA maturation, we inspected reads mapped to pre-tRNA, and observed a progressive increase in D20 levels during maturation for pre-tRNAs with U20a, while pre-tRNAs containing D20a did not exhibit such an increase. These results suggest that D20a occurs prior to D20 and suppresses its subsequent formation, supporting D20a as a cis-negative regulator of D20 installation.

We have included the discussion in the manuscript as follows: 'Pre-tRNA transcripts containing unmodified U20a exhibited a progressive increase in D20 levels during tRNA maturation, whereas transcripts with pre-existing D20a showed minimal changes (**Extended Data Fig. 7b-7d**). These findings, revealed by quantitative CRACI, suggest that D20a can be installed prior to D20 and acts as a cis-negative regulator of D20 formation, but not vice versa (**Fig. 4c**).'

Extended Data Figure 7. D dynamics characterized by CRACI. **b.** Schematic of the mapping methods for distinguishing pre-tRNAs and mature tRNAs. Reads containing additional nucleotides upstream of the annotated 5' end of the tRNA were classified as originating from pre-tRNAs, while reads that start precisely at the annotated 5' end were classified as mature tRNA reads. **c.** Comparison of mutation ratios at position 20a in tRNA-Gln (**c**) and tRNA-Ser (**d**) between mature tRNAs and pre-tRNAs. D20 levels increased during maturation in transcripts starting with U20a (blue), but remained stable in those with pre-existing D20a (gray). Each point represents one biological replicate.

4. *Some figure references in the main text are incorrect. Line 174, Fig 2h should be Fig 2i. Line 179, Fig 2i,j should be Fig 2j,k.*

Response: We apologize for these mistakes. We have corrected these mistakes. Thank you!

Reviewer #2: *The article by Cheng-Wei Ju and collaborators introduces the development and application of an innovative method named ‘Chemical Reduction Assisted Cytosine Incorporation sequencing’ (CRACI). This approach aims to quantitatively and comprehensively map dihydrouridylation sites within the transcriptome. The methodology relies on a chemical pre-reduction of the dihydrouracil (D) base in a given RNA molecule to tetrahydrouridine using potassium borohydride (KBH₄). In the presence of high GTP concentrations, this reduction enables HIV reverse transcriptase to incorporate a guanine (G) opposite tetrahydrouridine, leading to a detectable misincorporation during high-throughput sequencing.*

*This technique was applied to total RNA from human HepG2 cells, mouse embryonic stem cells (mESCs), and the model plant *Arabidopsis thaliana* to establish a detailed mapping of dihydrouridylation sites in cytosolic, mitochondrial, and chloroplast RNAs. The results confirmed expected observations based on previous tRNA sequencing data, such as: (i) The presence of D in cytosolic tRNAs at canonical dihydrouridylation positions (16, 17, 20, 20a, and 47); (ii) The localization of D20 in mammalian mitochondrial tRNAs. In parallel, the study unveiled novel and quite interesting equally important findings on the metabolism of dihydrouracil in tRNAs: (i) The presence of D at non-canonical positions (16 and 17) in mitochondrial tRNAs from HepG2 cells. (ii) A negative dependency between D20a and D20, suggesting an interdependence between these modifications. (iii) The coexistence of D16, D17, D20, and D20a in mitochondrial tRNAs of *A. thaliana* and D20 and D20a in chloroplast tRNAs. Using siRNA targeting the *Dus* enzymes (dihydrouridine synthases), the authors identified site-specificities for these enzymes. Most of the observations were consistent with known data, except for *Dus2*, which demonstrated a remarkable expanded specificity in a specific cellular context (HepG2 cells). In addition to its known role in depositing D20, *Dus2* was shown to introduce D16 and D17, a plasticity not observed in mESCs.*

*Beyond tRNAs, CRACI enabled exploration of dihydrouridylation in mRNAs, a controversial issue, confirming the presence of D in these RNAs, albeit at very low levels and in a limited number of transcripts. However, this technique does not detect dihydrouridylation sites in other types of non-coding RNAs, unlike other methodologies such as D-seq. Finally, a particularly innovative finding is the identification of D2467 in the 23S chloroplast rRNA of *A. thaliana*, marking the first report of this modification in eukaryotes, although it has been described in bacterial rRNAs.*

These results represent a significant advancement in the study of RNA dihydrouridylation. CRACI stands out from existing methods due to its specificity, quantitative nature, and the absence of abasic sites induced during the process. The conclusions are compelling and mark a major progress in a field that remains underexplored, despite the abundance of the D base in the transcriptome, second only to pseudouridine. This method opens promising new avenues for investigating dihydrouridylation

metabolism under physiological and pathological conditions. Consequently, I believe this article deserves publication in *Nature Communications* given the importance of the results obtained.

Response: We would like to thank the reviewer for the careful review and very positive comments. We have revised the manuscript according to the reviewer's valuable suggestions as detailed below.

Important comments for improvement:

1. *Absence of D20b:* The authors did not detect D20b in cytosolic tRNAs, although its existence has been reported. Do they have an explanation? This could reveal a limitation of the method, particularly for certain RNA types or specific sites, and should be mentioned in the revised manuscript.

Response: The previously reported D20b sites were identified in *S. cerevisiae*, primarily in Leu- and Tyr-tRNAs (ref: 10.1074/jbc.M401221200). However, such sites are extremely rare in *Mus musculus*, *Homo sapiens*, and *Arabidopsis thaliana*. Based on current database annotations, in these 3 species, D20b sites are only present in certain Leu-tRNAs, while Tyr-tRNAs do not possess any 20b positions in their tRNA sequence.

We carefully re-analyzed our data and confirmed the absence of D modifications at position 20b of annotated Leu-tRNAs in both *Mus musculus* and *Homo sapiens*. In *Arabidopsis thaliana*, in untreated 'input', we observed mutation signal at position 20b of Leu-tRNAs, suggesting the presence of other uridine modifications, but not D.

In *Arabidopsis thaliana*, D20b may exist in mitochondria- or chloroplast-encoded tRNAs. Given the lack of high-quality alignment for these organellar tRNAs, we provided tRNA sequences and the observed D sites within these tRNAs in Supplementary Table 6.

We have revised the manuscript to clarify these points. 'Although D20b sites have been reported in *S. cerevisiae* Leu and Tyr tRNAs¹¹, we did not detect any D20b in cytosolic tRNAs of *Mus musculus*, *Homo sapiens*, or *Arabidopsis thaliana*. Tyr tRNAs in these species lack a 20b position in their sequences (**Extended Data Fig. 10a-d**).^{56,57} While certain Leu tRNAs in *M. musculus* and *H. sapiens* do contain a 20b position, CRACI data confirmed the absence of D at these sites (**Extended Data Fig. 10e**). In *Arabidopsis thaliana*, a mutation signal was observed at Leu 20b in untreated input samples, indicating the possible presence of other uridine modifications, but not D'

(Due to large image dimensions, please refer to the original figure for full detail)

Extended Data Figure 10. D profiles at tRNA position 20b and in HepG2 rRNA. a–d, Isotype-specific consensus sequence features of tRNAs in *Saccharomyces cerevisiae* (a), *Homo sapiens* (b), *Mus musculus* (c), and *Arabidopsis thaliana* (d), obtained from the tRNAviz database⁵⁶. Potentially abundant U20b sites are marked with red circles. **e**. The T→C misincorporation ratio at U sites in multiple tRNA-

Leu isodecoders (TAG, TAA, CAG, AAG, and CAA) from HepG2 cells. The D fraction of each site was calculated as the average of two biological replicates.

2. *KBH₄ treatment conditions: The reduction of dihydrouridine to tetrahydrouridine depends on pH. The authors should specify the pH conditions used, as under basic pH, hydrolysis of the dihydrouracil heterocycle could produce ureidopropional compound, which undergoes β -elimination, resulting in an abasic site.*

Response: We have included detailed KBH₄ treatment conditions in Methods and measured the pH under our experimental setup, which is around pH 7.5. We further conducted a small-molecule reduction assay to confirm the formation of tetrahydrouridine. We have revised the manuscript as follows: ‘Liquid chromatography-tandem mass spectrometry (LC-MS/MS) analysis confirmed the near-quantitative formation of tetrahydrouridine, without detectable ureidopropional products under our KBH₄ reduction treatment condition (Extended Data Fig. 1a).’

Extended Data Figure 1. Optimization of RT conditions in CRACI. a. Model reaction of the reduction from the D nucleoside to THU using KBH₄ at 25 °C for 3 hours. Liquid chromatography-tandem mass spectrometry (LC-MS/MS) chromatograms comparing the retention times of standard THU and D with products from the reduction reactions.

3. *Orthogonal validation: While the results are robust, it would be valuable to confirm some findings using an independent method, such as MALDI-MS, particularly the presence of D16 and D17 in mitochondrial tRNAs from HepG2 cells. I understand this might be challenging due to the low quantities of mitochondrial tRNAs and the low stoichiometry of D, but it would be worth attempting.*

Response: This is a very good suggestion. We attempted MALDI-MS but were unable to detect corresponding signals due to the low abundance of mitochondrial tRNAs and the low stoichiometry of D in mt-tRNAs. Therefore, we used the antisense oligonucleotide (ASO)-based pulldown to enrich specific mt-tRNAs, followed by targeted quantification using LC-MS/MS quantification. We observed higher D/A ratios in mt-tRNA-Gln and mt-tRNA-Asn compared to mt-tRNA-Leu (UUR), which support our sequencing data. We have included this discussion into the manuscript as follows: ‘LC-MS/MS analysis of mt-tRNAs enriched by pulldown using DNA probes further supported these newly identified mitochondrial D sites

(Extended Data Fig. 4b).'

Extended Data Figure 4. D is absent in human small non-coding RNAs other than tRNA. b. Quantification of the D/A ratio in mitochondrial tRNAs (mt-tRNA-Pro, mt-tRNA-Lys, mt-tRNA-Leu(UUR), mt-tRNA-Asn, mt-tRNA-Gln) relative to mitochondrial small RNAs by LC-MS/MS. Probe-only samples were used as control.

4. *Integration of recent work: The authors should reference the recent study on hDus1L (Matsuura J et al., Commun Biol. 2024 Oct 2;7(1):1238. doi: 10.1038/s42003-024-06942-8).*

Response: We thank the reviewer for pointing this out. We have included this reference in the introduction part: 'A recent study identified DUS1L as the dihydrouridine synthase responsible for D16/D17 in human tRNAs, where DUS1L overexpression impairs tRNA processing and translation in glioblastoma.¹⁸'

5. Minor points:

Regarding functional redundancy among human Dus enzymes, the genes were not fully deleted and it is hard to judge. Recently, functional redundancy has been seen in Dus from B. subtilis but not in E. coli. Could the authors comment on this aspect, particularly how siRNA targeting induces compensation in D levels and which sites?

Response: We thank the reviewer for the suggestions. We have discussed the potential functional redundancy among human DUS enzymes and have included the following in the manuscript: 'At least 70% of D sites exhibited reduced modification levels following simultaneous knockdown of all four DUS enzymes (Extended Data Fig. 5f). However, because we performed siRNA-based transient knockdown, residual DUS protein and the long half-life of tRNAs may have allowed certain D sites to remain at high modification levels.³⁹ Thus, we cannot exclude the possibility of functional redundancy among human DUS enzymes.'

Extended Data Figure 5. CRAC1 captured reduced D levels in tRNAs upon DUS depletion in HepG2 cells. f. Scatter plot showing the correlation of mutation ratios between control cells

(siC) and combined *DUS1L/2L/3L/4L* knockdown (all DUS KD). Each dot represents a D site. Sites with significantly decreased mutation ratios upon DUS knockdown were shown in blue; non-significant sites in gray; sites with increased ratios in red. At least 70% of D sites exhibited reduced modification levels following DUS depletion. The mutation ratios for each point were calculated as the average of two biological replicates.

Line 50, DUS enzymes are not synthetases but synthases (dihydrouridine synthetases). Please make the correction. By convention synthetases are enzymes that employ ATP in the catalyzed reaction like for instance amino-acyl tRNA synthetases.

Line 50: Reference 8 discusses kinetic studies on a single Dus enzyme, Dus2p, but does not address the issue of selectivity between NADPH and NADH. While Dus enzymes preferentially use NADPH, they can also utilize NADH, albeit with a lower KM . This is clarified in Reference 2.

Line 177: Replace 'mass spec' with 'mass spectrometry.'

Line 285: Correct 'DUC' to 'DUS.'

Response: We sincerely thank the reviewers for all these comments. We have revised the manuscript and fixed all these points.

Line 51: Reference 9 is not cited correctly. This work does not discuss DusA, DusB, or DusC. Furthermore, most bacteria do not possess all three Dus enzymes. The presence of all three Dus enzymes is specific to Proteobacteria. In contrast, as noted in Reference 10, Gram-positive bacteria possess only DusB and lack both DusA and DusC.

Response: We thank the review for the careful review. We have revised the manuscript and made the correction: 'The presence of all three Dus enzymes (DusA, DusB, DusC) were observed in Proteobacteria, while Gram-positive bacteria only possess DusB.^{9,10}'

Reference:

9. Faivre, B. et al. Dihydrouridine synthesis in tRNAs is under reductive evolution in Mollicutes. *RNA Biology* **18**, 2278–2289 (2021).

10. Kasprzak, J. M., Czerwoniec, A. & Bujnicki, J. M. Molecular evolution of dihydrouridine synthases. *BMC Bioinformatics* **13**, 153 (2012).

Reviewer #3: *This paper co-led by Li-Sheng Zhang and Chuan He describes new Chemical Reduction Assisted Cytosine Incorporation seq approach, abbreviated CRACI, which maps D modifications in the RNAome. This approach utilizes chemical reduction with KBH₄, previously applied to N⁴-acetylcytidine, to reduce D. It is followed by reverse transcriptase reactions that induce mutations at RNA modification sites in the presence of elevated GTP (1 mM) to dNTP (10 μM) ratios, yielding a reliable sequencing method for identifying D modifications.*

*The D modification is an abundant feature in tRNAs, which are the primary species detected using CRACI. Through siRNA-based silencing of individual DUS enzymes, the authors comprehensively map tRNA modifications of both cytoplasmic and mitochondrial origin in two eukaryotic cell lines, uncovering several new modification sites in mito-tRNAs. They also apply the method to *A. thaliana* seedlings, revealing a similar distribution of D modifications between cytoplasmic tRNAs of mammals and plants, but notable differences in mito-tRNAs. *A. thaliana* mito-tRNAs are more extensively modified than their mammalian counterparts. Ribosomal RNA and ncRNAs lack D modifications, except for chloroplast 23S rRNA in *A. thaliana*, which resembles bacterial 23S D2449. Although a few low-level modifications were detected in human mRNA, their origin remains unclear.*

In summary, this is well-written manuscript with a straight forward experimental design and convincing results. However, a few aspects require clarification before considering the manuscript for publication:

Response: We would like to thank the reviewer for the very positive comments. We have revised the manuscript according to the reviewer's valuable suggestions as detailed below.

1. *In the section describing the quantification of the D stoichiometry (from l. 131 on), it is unclear why 5-nt motif has been considered. Some reports (<https://pmc.ncbi.nlm.nih.gov/articles/PMC10635142/>) show that motifs as small as two nucleotides (GU) could be recognized by DUS. The authors should explain their rationale for selecting the motifs and the filtering scheme.*

Response: We thank the reviewer for the comments. We selected a 5-nt motif centered by U or D to provide 256 different motifs, allowing us to better assess the general compatibility of the RT enzyme used in CRACI across different sequence contexts. For example, SSIV RT shows poor read-through when double uridines are located following D, while HIV RT performs well under nearly all sequences. This 5-nt probe helps evaluate potential sequence-dependent biases of the reverse transcriptases tested here. We have added a clarification in the revised manuscript. 'This indicates the use of HIV-RT in CRACI does not lead to significant sequence context bias around D sites (Fig. 1d).'

2. *The sequencing protocol used in this study is not fundamentally new, and the original protocols should be cited. The sequencing approach with direct adaptor ligation was originally developed for miRNA and now is widely employed in ribosome profiling (Ribo-seq) protocols. The original publications should be appropriately acknowledged.*

Response: We thank the reviewer for the comments. We agree and do not wish to overstate the sequencing protocol. We have revised the manuscript accordingly and cited the original publications as suggested: *'This strategy of adaptor ligation combined with UMI has been broadly applied in previous studies of small RNAs and miRNAs, showing low bias, high reproducibility, and effective PCR duplicate removal^{41,42.}'*

3. *Since this is a methodological paper, it should provide more explanation on how in a polyA-selection step small RNAs are retained. Are small RNAs quantitatively retained? Why polyA selection is chosen over rRNA depletion? The first part of the paper would definitely benefit from more through explanation of the steps supported by evidence on how quantitative each fraction is?*

Response: We thank the reviewer for the comments. We actually prepared two types of cellular RNA: (1) small RNAs (<200 nt) isolation through size selection; (2) polyA⁺ RNA enrichment through oligo-dT pulldown. polyA⁺ RNA enrichment allowed us to obtain much improved read depth for mRNAs and lncRNAs, compared to using total RNA directly, enabling CRACI to detect D sites in low-abundance transcripts. We additionally tested rRNA depletion using the RiboMinus™ Kits from ThermoFisher; however, CRACI did not observe differences in detected DHU sites compared to using polyA enrichment, and therefore we did not include the data. RiboMinus™ Kits are also more expensive than polyA⁺ RNA purification kits.

To clarify the 'quantitative' feature, in CRACI, we emphasize the estimation of D modification stoichiometry, in the format of (DHU / [DHU + U]) ratio, rather than the absolute abundance of RNA species.

We have included these details in Methods section.

4.1. *In the section 'Quantitative CRACI uncovers cytoplasmic and mitochondrial tRNA' (l. 150-171), the authors detect different number of D modifications in the tRNAs, with some having only a single modification. A correlation with tRNA abundance would be helpful to demonstrate that a lower number of modifications is not associated with low-abundance tRNAs.*

Response: We thank the reviewer for the comments. We included more analysis in the revised manuscript as follows: *'The absence of correlation between tRNA expression levels and detected D sites*

confirms that the observed D modifications are unlikely affected by sequencing coverage (**Extended Data Fig. 3c**).'

Extended Data Figure 3. D profiles in HepG2 ct-tRNA and mt-tRNA. c.
Relationship between the number of D sites and the expression level of tRNAs

4.2. Additionally, the authors should compare the D modification map generated by CRACI with previously described D positions in mammalian tRNAs, as documented in Modomics and GtRNAdb.

Response: We thank the reviewer for the comments. When conducting the comparison of D modifications detected by CRACI with previously reported mammalian tRNA D sites, we noticed that the reliable assignment of D modifications in previously published databases is limited.

The information in GtRNAdb is obtained from Modomics, we therefore focused on the curated dataset in Modomics. Notably, a majority of D sites in Modomics are labeled with a confidence score of 5, meaning 'Evidence not yet annotated (Unknown).' Our CRACI detected 27 out of 30 D sites annotated in Modomics, demonstrating a high level of concordance.

The three Modomics sites not detected by CRACI are:

- tRNA-His-GTG 16: We confirmed the absence of a detectable D modification at this site, at least in HepG2 cell line.
- tRNA-Met-CAT 20: Although a high mutation ratio was seen in CRACI-treated sample, this site was not collected as a D site due to the same high mutation ratio in untreated input sample, suggesting the presence of other uridine modification such as acp³U or acp³D, which can disrupt base pairing during RT and directly cause mutations even without CRACI treatment.
- tRNA-Tyr-GTA-16: Similarly, high mutation ratios were observed in both CRACI-treated and untreated input samples, leading to exclusion during D site calling. Again, this may be attributed to the presence of other uridine modification at this position.

We have included these discussion in the revised manuscript as follows:

Additionally, the overlap between our data and Modomics further supports the reliability of CRACI-based D detection.⁴⁸ Although Modomics annotated only 30 D sites, most with low confidence (score 5, indicating "Evidence not yet annotated [Unknown]"), CRACI successfully detected and verified 27 of them (**Extended Data Fig. 3i**). This strong concordance underscores the robustness and sensitivity of CRACI. The three Modomics sites not detected by CRACI are detailed in **Extended Data Fig. 3j**: (1) tRNA-His-GTG D16, where no detectable D modification was observed in HepG2 cells; (2) tRNA-Met-CAT D20, where high mutation ratios were seen in both CRACI-treated and input samples, likely due to background

mutations from other uridine modifications such as acp3U or acp3D; and (3) tRNA-Tyr-GTA D16, with similar high background mutation ratios possibly caused by non-D modifications. Together, these findings demonstrate CRACI's high accuracy and confidence in transcriptome-wide D detection.

Extended Data Figure 3. D profiles in HepG2 ct-tRNA and mt-tRNA. i. Table of D sites recorded in Modomics across different tRNAs. Green boxes indicate sites detected by CRACI, while yellow boxes indicate sites not detected by CRACI. **j.** The T→C misincorporation ratio at U sites in tRNA-Met-CAT-1-1 tRNA-Tyr-GTA-2-1, and tRNA-His-GTG-1-1 in HepG2. Highlighted regions correspond to Modomics recorded D sites. The mutation ratios are calculated as the average of two biological replicates.

4.3. In the same paragraph, which are the tRNAs outside the 10 top matches, that do not overlap with the CLIP/iCLIP data? A list of those would be helpful.

Response: We thank the reviewer for the comment. We have included a list to show these tRNAs, and added a comment in the revised manuscript: ‘CRACI also identified several novel D sites not captured by CLIP experiments (Extended Data Fig. 3f-h).’

Extended Data Figure 3. D profiles in HepG2 ct-tRNA and mt-tRNA. f. Overlap between CRACI-detected D16/17 sites and 5-CIUrd-iCLIP targets of DUS1L. **g.** Overlap between CRACI-detected D20 sites and 5-CIUrd-iCLIP targets of DUS2L. **h.** Overlap between CRACI-detected D47 sites and 5-CIUrd-iCLIP targets of DUS3L.

5.1. In the paragraph 'CRACI assigns the 'writer' proteins to D modifications in human cytoplasmic and mitochondrial tRNAs' (l. 185-201), the levels of DUS enzymes were silenced by siRNA to different extent. The expression levels of DUS1L and DUS4L were particularly low; however, the stoichiometry of the corresponding D positions remains fairly high. Only a few tRNAs appear sensitive to the absence of these enzymes, while for the majority, the modification levels remain at nearly 100% (Extended 4 and 3b-e). This unexpected result suggests possible cross-reactivity among DUS enzymes. A critical control is missing here: the authors should silence all DUS enzymes together and demonstrate that D modifications are nearly absent (<https://academic.oup.com/nar/article/52/21/12784/7845166>). This would serve as an important control to validate the sensitivity of CRACI.

Response: We thank the reviewer for the comment. Our current manuscript involved siRNA-mediated transient knockdown of individual DUS enzyme. Although transcript levels of these DUS enzymes were significantly reduced after siRNA knockdown, residual protein may still be present during the 72-hour time window, contributing to the preserved D modification levels at certain sites. Note that lifetime of some tRNAs can be even several days, resulting in the persistence of the D modification as well. Similar observations have been reported in other studies on tRNA and rRNA modification (e.g., FBL-mediated Nm methylation uncovered by Nm mut-seq, ref: 10.1038/s41422-023-00836-w), where the loss of 'writer' protein does not completely abolish tRNA modification due to the stability of pre-existing ones.

To address the reviewer's concern, we performed a combined siRNA knockdown with simultaneously adding siRNAs targeting *DUS1L*, *DUS2L*, *DUS3L*, and *DUS4L*. This one-pot DUS enzyme depletion led to a substantial reduction in D modification levels across most tRNAs, as shown in Extended Data Fig. 4f, validating the sensitivity of CRACI. Above 70% of D sites exhibited reduced modification levels following simultaneous knockdown of all four DUS enzymes. Nonetheless, since this approach still relies on transient knockdown, residual DUS protein and the inherent stability of mature tRNAs likely account for the incomplete loss of some D sites.

We have included in the revised manuscript as follows: 'At least 70% of D sites exhibited reduced modification levels following simultaneous knockdown of all four DUS enzymes (**Extended Data Fig. 5f**). However, because we performed siRNA-based transient knockdown, residual DUS protein and the long half-life of tRNAs may have allowed certain D sites to remain at high modification levels.³⁹ Thus, we cannot exclude the possibility of functional redundancy among human DUS enzymes.'

Extended Data Figure 5. CRACI captured reduced D levels in tRNAs upon DUS depletion in HepG2 cells. f. Scatter plot showing the correlation of mutation ratios between control cells (siC) and combined *DUS1L/2L/3L/4L* knockdown (all DUS KD). Each dot represents a D site. Sites with significantly decreased mutation ratios upon DUS knockdown were shown in blue; non-significant sites in gray; sites with increased ratios in red. At least 70% of D sites exhibited reduced modification levels following DUS depletion. The mutation ratios for each point were calculated as the average of two biological replicates.

5.2. In the same paragraph, the claim that the study establishes ‘...DUS2L as the sole active ‘writer’ protein responsible for D modifications in HepG2 mitochondria’ should be softened, as the data do not clearly support this strong claim.

Response: We thank the reviewer for the comment. We have revised the manuscript to avoid overclaim. ‘This finding validated newly identified D sites and suggested DUS2L as an active ‘writer’ protein responsible for D modifications in HepG2 mitochondria (Fig. 3f,g).’

6. The conclusion in l. 220-221 regarding the secondary structure is somewhat overstated, as there is no direct evidence supporting the involvement of these regions in a secondary structure. Alternatively, this could simply reflect differences in enzymatic activities (<https://academic.oup.com/nar/article/52/21/12784/7845166>) - a scenario that the authors should consider in their discussion.

Response: We thank the reviewer for the comment. We have revised the manuscript accordingly: ‘This inhibitory effect may stem from D modification-induced changes in local RNA structure, which could affect DUS enzyme recruitment. However, we cannot exclude the possibility that differences in enzymatic activity among DUS variants also contribute.⁵¹’

7. The D sites in human mRNA are sparse. Have those 8 sites been found in previous studies? What is the reproducibility of detecting these sites across biological replicates? Additionally, the conclusion regarding potential secondary structures seems too strong and lacks experimental evidence (e.g., structure-seq data). The authors should tone it down and also discuss alternative possibilities. It cannot be ruled out that D incorporation in mRNA is not enzymatic, and incorporated as pre-modified D nucleotide (e.g., originating from degradation of the abundant tRNA^{ome}) during mRNA synthesis.

Response: We thank the reviewer for the comment. Regarding the 8 D sites detected in mRNA, to our knowledge, these sites have not been reported in previous studies. However, we were able to reproducibly detect these sites across three biologically independent replicates, supporting their existence.

We agree with the reviewer and revised the manuscript to tone down the interpretation here involving secondary structures, since the low modification stoichiometry of these D sites makes it challenging to correlate with RNA structures. We have revised the manuscript and discussed the possible non-enzymatic possibilities for D generation as the follows: 'Given that D is typically installed within structured regions of tRNA, we speculate that some D deposition in mRNA may similarly be associated with local secondary structures. However, the observed low modification fraction at these mRNA D sites suggests that they could also arise through non-enzymatic processes, such as the salvage and incorporation of pre-modified D nucleotides derived from tRNA degradation.'

8. *The legend of each figure/panel should provide information on the number of biological replicates. There are many figures without SD. Do they represent merged replicates?*

Response: We thank the reviewer for the comment. We have revised the figure legends to include the number of biological replicates as suggested. For figures without SD, values represent the average of at least two biological replicates. The data values for all Figures and Extended Data Figures are provided in the Supplementary Tables.

Response to comments from reviewers

Reviewer #1: *The authors have addressed my comments in the revision.*

Response: We would like to once again thank the reviewer for helpful suggestions and insightful comments, which have greatly assisted us in improving the manuscript.

Reviewer #2: *The authors have convincingly addressed all of my concerns, and the paper now merits publication in Nature Communications. However, there are a few minor corrections that should be made before final acceptance:*

Response: We are grateful to the reviewer for constructive suggestions, which have helped us strengthen the manuscript.

Figure 1C: There are two labeling errors: "DMTP" should be corrected to "dNTP".

Response: We have revised accordingly.

Line 127: What does "pf" stand for? This should be corrected.

Response: We have revised accordingly.

Figure 2C: It would have been insightful to also include D17 and D20a for comparison.

Response: We thank the reviewer for the suggestion. We do include D17 in the comparison with the DUS1L target. However, for D20a, we could not identify high-quality metabolic labeling CLIP data, making it difficult to include at this stage.

*Line 271: References 9 and 50 do not support the claimed role of hDus2's dsRBD in tRNA structure recognition. The correct references are 44 and the following study: Biochemistry 2019, *58*(20), 2463–2473. DOI: 10.1021/acs.biochem.9b00111.*

Response: We have revised accordingly.

Reviewer #3: *With the addition of several new experiments and analyses, the authors have adequately addressed all concerns and comments from the previous round.*

Response: We sincerely appreciate the reviewer's constructive suggestions and insightful comments.